# Sleep prevents catastrophic forgetting in spiking neural networks by forming a joint synaptic weight representation

**Ryan Golden**[1,2☯], **Jean Erik Delanois**[2,3☯], **Pavel Sanda**[4], **Maxim Bazhenov**[1,2]*

**1** Neurosciences Graduate Program, University of California, San Diego, La Jolla, California, United States of America, **2** Department of Medicine, University of California, San Diego, La Jolla, California, United States of America, **3** Department of Computer Science and Engineering, University of California, San Diego, La Jolla, California, United States of America, **4** Institute of Computer Science of the Czech Academy of Sciences, Prague, Czech Republic

☯ These authors contributed equally to this work.
* mbazhenov@ucsd.edu

**Data Availability Statement:** All relevant data are within the paper and its Supporting Information files.

## Abstract

Artificial neural networks overwrite previously learned tasks when trained sequentially, a phenomenon known as catastrophic forgetting. In contrast, the brain learns continuously, and typically learns best when new training is interleaved with periods of sleep for memory consolidation. Here we used spiking network to study mechanisms behind catastrophic forgetting and the role of sleep in preventing it. The network could be trained to learn a complex foraging task but exhibited catastrophic forgetting when trained sequentially on different tasks. In synaptic weight space, new task training moved the synaptic weight configuration away from the manifold representing old task leading to forgetting. Interleaving new task training with periods of off-line reactivation, mimicking biological sleep, mitigated catastrophic forgetting by constraining the network synaptic weight state to the previously learned manifold, while allowing the weight configuration to converge towards the intersection of the manifolds representing old and new tasks. The study reveals a possible strategy of synaptic weights dynamics the brain applies during sleep to prevent forgetting and optimize learning.

## Author summary

Artificial neural networks can achieve superhuman performance in many domains. Despite these advances, these networks fail in sequential learning; they achieve optimal performance on newer tasks at the expense of performance on previously learned tasks. Humans and animals on the other hand have a remarkable ability to learn continuously and incorporate new data into their corpus of existing knowledge. Sleep has been hypothesized to play an important role in memory and learning by enabling spontaneous reactivation of previously learned memory patterns. Here we use a spiking neural network model, simulating sensory processing and reinforcement learning in animal brain, to demonstrate that interleaving new task training with sleep-like activity optimizes the

**Funding:** This study was supported by ONR (N00014-16-1-2829 to MB), Lifelong Learning Machines program from DARPA/MTO (HR0011-18-2-0021 to MB), NSF (EFRI BRAID 2223839 to MB), and NIH (1RF1MH117155 to MB; 1R01MH125557 to MB; 1R01NS109553 to MB). The funders had no role in study design, data collection and analysis, decision to publish, or preparation of the manuscript.

**Competing interests:** The authors have declared that no competing interests exist.

network's memory representation in synaptic weight space to prevent forgetting old memories. Sleep makes this possible by replaying old memory traces without the explicit usage of the old task data.

## Introduction

Humans are capable of continuously learning to perform novel tasks throughout life without interfering with their ability to perform previous tasks. Conversely, while modern artificial neural networks (ANNs) are capable of learning to perform complicated tasks, ANNs have difficulty learning multiple tasks sequentially [1–3]. Sequential training commonly results in catastrophic forgetting, a phenomenon which occurs when training on the new task completely overwrites the synaptic weights learned during the previous task, leaving the ANN incapable of performing a previous task [1–4]. Attempts to solve catastrophic forgetting have drawn on insights from the study of neurobiological learning, leading to the growth of neuroscience-inspired artificial intelligence (AI) [5–8]. While proposed approaches are capable of mitigating catastrophic forgetting in certain circumstances, a general solution which can achieve human level performance for continual learning is still an open question [9].

Historically, an interleaved training paradigm, where multiple tasks are presented within a common training dataset, has been employed to circumvent the issue of catastrophic forgetting [4,10,11]. In fact, interleaved training was originally construed to be an approximation to what the brain may be doing during sleep to consolidate memories; spontaneously reactivating memories from multiple interfering tasks in an interleaved manner [11]. Unfortunately, explicit use of interleaved training, in contrast to memory consolidation during biological sleep, imposes the stringent constraint that the original training data be perpetually stored for later use and combined with new data to retrain the network [1,2,4,11]. Thus, the challenge is to understand how the biological brain enables memory reactivation during sleep without access to past training data.

Parallel to the growth of neuroscience-inspired ANNs, there has been increasing investigation of spiking neural networks (SNNs) which attempt to provide a more realistic model of brain functioning by taking into account the underlying neural dynamics and by using biologically plausible local learning rules [12–15]. A potential advantage of the SNNs, that was explored in our new study, is that local learning rules combined with spike-based communication allow previously learned memory traces to reactivate spontaneously and modify synaptic weights without interference during off-line processing–sleep. Indeed, a common hypothesis, supported by a vast range of neuroscience data, is that the consolidation of memories during sleep occurs through synaptic changes enabled by reactivation of the neuron ensembles engaged during learning [16–20]. It has been suggested that Rapid Eye Movement (REM) sleep supports the consolidation of non-declarative or procedural memories, while non-REM sleep supports the consolidation of declarative memories [16,21–23].

Here we used a multi-layer SNN with reinforcement learning to investigate whether interleaving periods of new task training with periods of sleep-like autonomous activity, can circumvent catastrophic forgetting. The network can be trained to learn one of two complementary complex foraging tasks involving pattern discrimination but exhibits catastrophic forgetting when trained on the tasks sequentially. Significantly, we show that catastrophic forgetting can be prevented by periodically interrupting reinforcement learning on a new task with sleep-like phases. From the perspective of synaptic weight space, while new task training alone moves the synaptic weight configuration away from the old task's manifold–a subspace of synaptic weight space that guarantees high performance on that task—and towards

the new task manifold, interleaving new task training with sleep replay allows the synaptic weights to stay near the old task manifold and still move towards its intersection with the manifold representing the new task, i.e., converge to the intersection of these manifolds. Our study predicts that sleep prevents catastrophic forgetting in the brain by forming joint synaptic weight representations suitable for storing multiple memories.

## Results

Human and animal brains are complex and although there are many differences between species, critical common elements can still be identified from insects to humans. From an anatomic perspective, this includes largely the sequential processing of sensory information, from raw low level representations on the sensory periphery to high level representations deeper in the brain followed by decision making networks controlling the motor circuits. From a functional perspective, this includes local synaptic plasticity, combination of different plasticity rules and sleep-wake cycle that was shown to be critical for memory and learning in variety of species from insects [24–26] to vertebrates [16]. In this new study we model a basic brain neural circuit including many of these anatomical and functional elements. While our model is extremely simplified, it captures critical processing steps found, e.g., in insect olfactory system where odor information is sent from olfactory receptors to the mushroom bodies and then to the motor circuits. In vertebrates, visual information is sent from the retina to early visual cortex and then to decision making layers in associative cortices to drive motor output. Many of these steps are plastic, in particular decision making circuits utilize spike timing dependent plasticity (STDP) in insects [27] and vertebrates [28,29].

Fig 1A illustrates a feedforward spiking neural network (see also *Methods*: *Network Structure* for details) simulating the basic steps from sensory input to motor output. Excitatory synapses between the input (I) and hidden (H) layers were subjected to unsupervised learning (implemented as non-rewarded STDP) [28,29] while those between the H and output (O) layers were subjected to reinforcement learning (implemented using rewarded STDP) [30–33] (see *Methods*: *Synaptic plasticity* for details). Unsupervised plasticity allowed neurons in layer H to learn different particle patterns at various spatial locations of the input layer I, while rewarded STDP allowed the neurons in layer O to learn motor decisions based on the type of the particle patterns detected in the input layer [14]. While inspired by the processing steps of a biological brain, this structure also mimics basic elements of the feedforward artificial neural networks (ANNs), including convolutional layer (from I to H) and fully connected layer (from H to O) [34].

### Complementary complex foraging tasks can be robustly learned

We trained the network on one of two complementary complex foraging tasks. In either task, the network learned to discriminate between rewarded and punished particle patterns in order to acquire as much reward as possible. We consider pattern discriminability (ratio of rewarded vs punished particles consumed) as a measure of performance, with chance performance being 0.5. All reported results are based on at least 10 trials with different random network initialization.

The paradigm for Task 1 is shown in Fig 1B. First, during an unsupervised learning period, all 4 types of 2-particle patterns (horizontal, vertical, positive diagonal, and negative diagonal) were present in the environment with equal densities. This was a period, equivalent to a developmental critical period in the brain (or training convolutional layers in ANN), when the network learned the environmental statistics and formed, in layer H, high level representations of all possible patterns found at the different visual field locations (see Fig 2 for details).

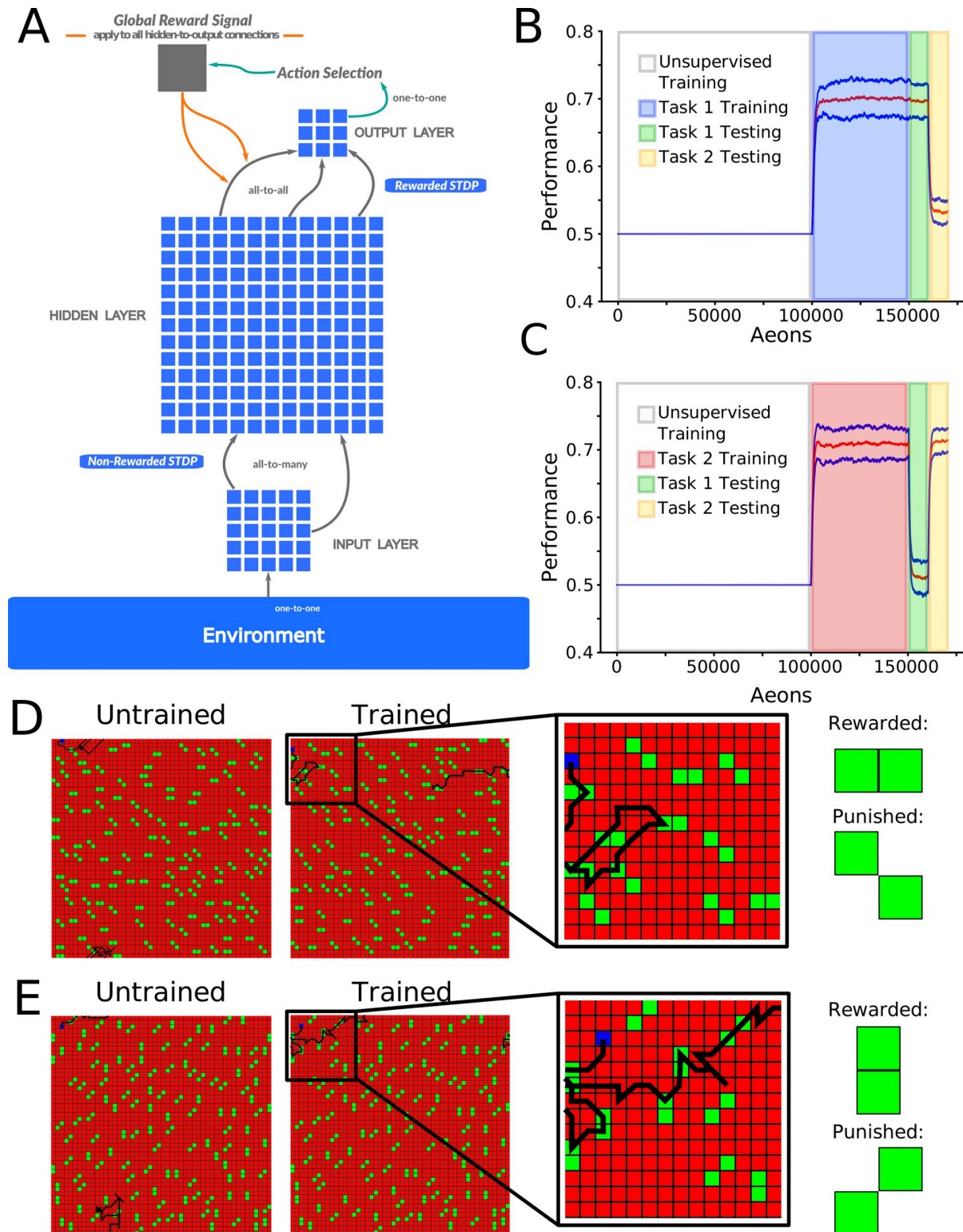

**Fig 1. Network architecture and foraging task structure. (A)** The network had three layers of neurons with a feed-forward connectivity scheme. Input from virtual environment was simulated as a set of excitatory inputs to the input layer neurons ("visual field"- 7x7 subspace of 50x50 environment) representing the position of food particles in an egocentric reference frame relative to the virtual agent. Each hidden layer neuron received an excitatory synapse from 9 randomly selected input layer neurons. Each output layer neuron received one excitatory and one inhibitory synapse from each hidden layer neuron. The most active neuron in the output layer (size 3x3) determined the direction of

movement. **(B)** Mean performance (redline) and standard deviation (blue lines) over time: unsupervised training (white), Task 1 training (blue), and Task 1 (green) and Task 2 (yellow) testing. The y-axis represents the agent's performance, or the probability of acquiring rewarded as opposed to punished particle patterns. The x-axis is time in aeons (1 aeon = 100 movement cycles). **(C)** The same as shown in (B) except now for: unsupervised training (white), Task 2 training (red), and Task 1 (green) and Task 2 (yellow) testing. **(D)** Examples of trajectories through the environment at the beginning (left) and at the end (middle-left) of training on Task 1, with a zoom in on the trajectory at the end of training (middle-right), and the values of the task-relevant food particles (right). **(E)**. The same as shown in (D) except for Task 2.

Unsupervised training was followed by a reinforcement learning period, equivalent to task specific training in the brain (or training a specific set of classes in an ANN), during which the synapses between layers I and H were frozen while synapses from H to O were updated using a

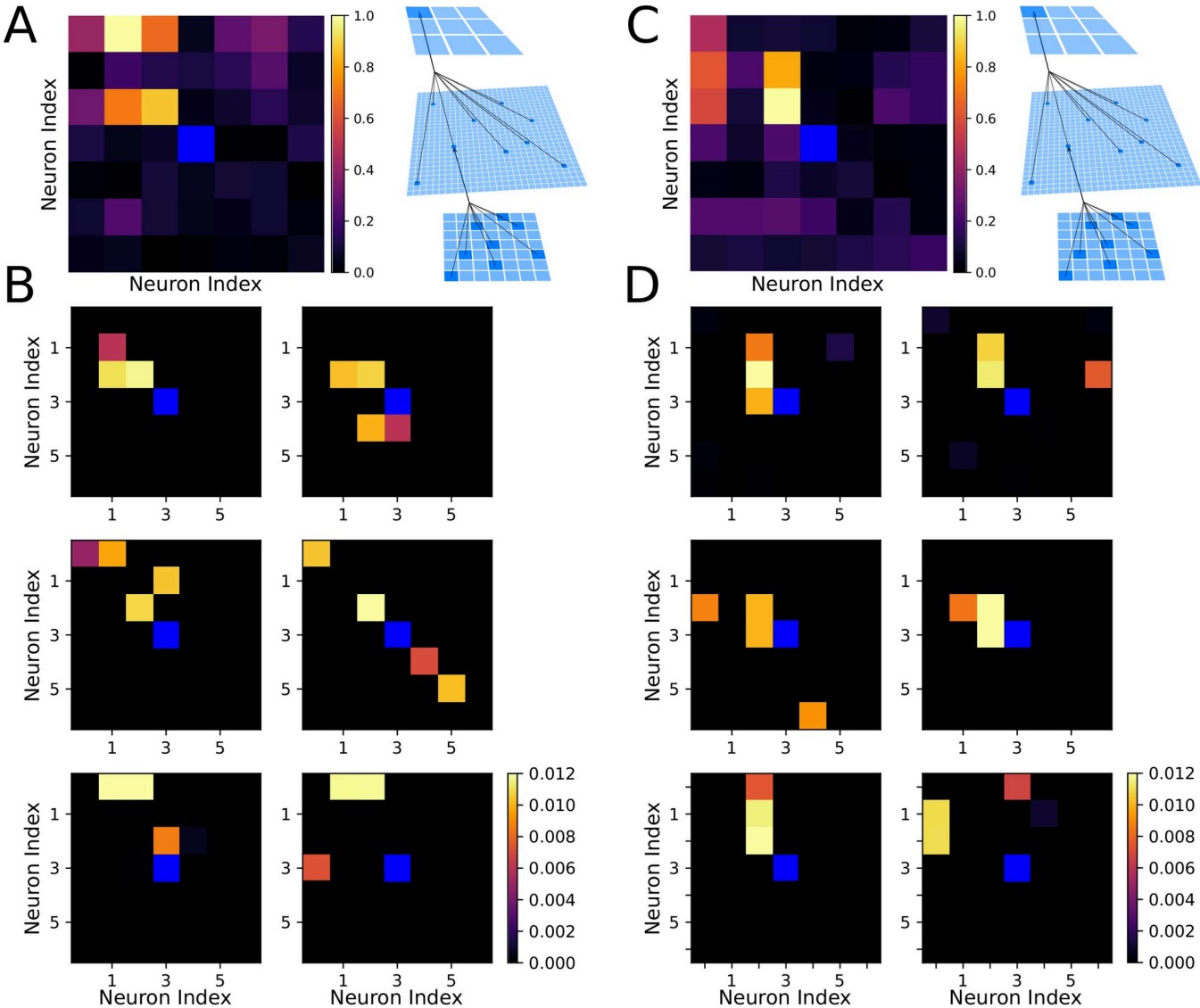

**Fig 2. Receptive fields of output and hidden layer neurons determine the agent behavior. (A)** Left, Receptive field of the output layer neuron controlling movement to the upper-left direction following training on Task 1. This neuron can be seen to selectively respond to horizontal orientations in the upper-left quadrant of the visual field. Right, Schematic of connections between layers. **(B)** Examples of receptive fields of hidden layer neurons which synapse strongly onto the output neuron from (A) after training on Task 1. **(C)** The same as shown in (A) except following training on Task 2. The upper-left decision neuron can be seen to selectively respond to vertical orientations in the upper-left quadrant of the visual field. **(D)** The same as shown in (B) except following training on Task 2.

rewarded STDP rule. The reinforcement learning period was when the network learned to make decisions about which direction to move based on the visual input. For Task 1, horizontal patterns were rewarded and negative diagonal patterns were punished (Fig 1D). During both the rewarded training and the testing periods only 2 types of patterns were present in the environment (e.g. horizontal and negative diagonal for Task 1).

After training Task 1, mean performance across ten trials on Task 1 was 0.70 ± 0.02 while performance on the untrained Task 2 was 0.53 ± 0.02 (chance level). The naive agent moved randomly through the environment (Fig 1D, left), but after task training, moved to seek out horizontal patterns and largely avoid negative diagonal ones (Fig 1D, right). The complementary paradigm for Task 2 (vertical patterns are rewarded, and positive diagonal are punished) is shown in Fig 1C and 1E. These results demonstrate that the network is capable of learning and performing either one of the two complementary complex foraging tasks. The similarity between these tasks is evident in their definition (symmetrical particle orientations; Fig 1D and 1E), through the similar performances attained by the network on each task (Fig 1B and 1C), and through the similar levels of activity induced in the network when training each task (S1A and S1B Fig)

To understand how sensitive a trained network was to pruning, we employed a neuronal dropout procedure which progressively removes neurons from the hidden layer at random (S2 Fig). We found the network was able to keep performance steady on either task following training until around 70% of the hidden layer was pruned. Such high resiliency suggests the network utilizes a highly distributed coding strategy to develop its policy.

Next, to understand synaptic changes during training, we computed receptive fields of each neuron in layer O with respect to the inputs from layer I (see schematic in Fig 2A and 2C). This was done by first computing the receptive fields of all of the neurons in layer H with respect to I, then performing a weighted average where the weights were given by the synaptic strength from each neuron in layer H to the particular neuron in layer O. Fig 2A shows a representative example of the receptive field which developed after training on Task 1 for one specific neuron in layer O which controls movements to the upper-left direction. This neuron responded most robustly to bars of horizontal orientation (rewarded) in the upper-left quadrant of the visual field and, importantly, did not respond to bars of negative diagonal orientation (punished).

Fig 2B shows examples of receptive fields of six neurons in layer H which synapse strongly onto the upper-left neuron in layer O (the neuron shown in Fig 2A). These neurons formed high level representations of the input patterns, similar to the neurons in the primary visual system or later layers of a convolutional neural network [35–37]. The majority of these receptive fields revealed strong selection for the horizontal (i.e. rewarded) food particles in the upper-left quadrant of the visual field. As a particularly notable example, one of these layer H neurons (Fig 2B; middle-right) preferentially responded to negative diagonal (i.e. punished) food particles in the bottom-right quadrant of the visual field. Thus, spiking in this neuron caused the agent to move away from these punished food particles. Similar findings after training on Task 2 are shown in Fig 2C and 2D.

To further quantify the network's sensitivity to various particle types we developed a metric termed the Particle Responsiveness Metric (PRM) to gauge how specific particles influence activity of the output layer neurons (see the section Methods: Particle responsiveness metric for further details). Using PRM on all food particle orientations across ten trials, we found that following Task 1 training the network is drawn to horizontal particles (S3A Fig) while post Task 2 training vertical particles drive output layer activity (S3B Fig), thus quantitatively supporting the qualitative results displayed in Fig 2.

## Sleep prevents catastrophic forgetting of the old task during new task training

We next tested whether the model exhibits catastrophic forgetting by training sequentially on Task 1 (old task) followed by Task 2 (new task) (Fig 3A). Following Task 2 training, mean performance across ten trials on Task 1 was down to no better than chance (0.52 ± 0.02), while performance on Task 2 improved to 0.69 ± 0.03 (Fig 3A and 3B). Thus, sequential training on a complementary task caused the network to undergo catastrophic forgetting of the task trained earlier, remembering only the most recent task.

Interleaved training was proposed as a solution for catastrophic forgetting [4,10,11]. In the next experiment, after training on Task 1, we simulated interleaved T1/T2 training (Interleaved$_{T1,T2}$) when we alternated short presentations of Task 1 and Task 2 every 100 movement cycles (Fig 3C). Sample network activity from this period can be seen to closely resemble single task training (S1C Fig). Following interleaved training, the network achieved a mean performance of 0.68 ±0.03 on Task 1 and a performance of 0.65 ± 0.04 on Task 2 across trials. Therefore, interleaved training allowed the network to learn new Task 2 without forgetting previously learned Task 1. However, while interleaved training made it possible to learn both tasks, it imposes the stringent constraint that all the original training data (in our case explicit access to the Task 1 environment) be stored for later use and combined with new data to retrain the network [1,2,4,11].

Sleep is believed to be an off-line processing period when recent memories are replayed to avoid damage from new learning. We previously showed that sleep replay improves memory in a thalamocortical network [38–40] and when a network was trained to learn interfering tasks sequentially, sleep prevented the old task memory from catastrophic forgetting [41]. Can we implement a sleep like phase to our model to protect an old task and still accomplish new task learning without explicit re-training of the old task? In vivo, activity of the neocortical neurons during REM sleep is low-synchronized and similar to baseline awake activity [42]. Therefore, to simulate REM sleep-like activity in the model, the rewarded STDP rule was replaced by unsupervised STDP, the input layer was silenced while hidden layer neurons were artificially stimulated by Poisson distributed spike trains in order to maintain spiking rates similar to that during task training (see *Methods*: *Simulated Sleep* for details). Sample network activity recorded during this sleep phase is visualized in the raster plots shown in S1D Fig.

Again, we first trained the network on Task 1. Next, we implemented a training phase consisted of alternating periods of training on Task 2 (new task) lasting 100 movement cycles and periods of "sleep" of the same duration (we will refer to this training phase as Interleaved$_{S,T2}$) (Fig 3E). Importantly, no training on Task 1 was performed at any time during Interleaved$_{S,T2}$. Following Interleaved$_{S,T2}$, the network achieved a mean performance across ten trials of 0.68 ± 0.05 on Task 2 and retained a performance of 0.70 ± 0.03 on Task 1 (Fig 3E and 3F), comparable to single Task 1 (0.70 ± 002) or Task 2 (0.69 ± 0.03) performances (Fig 1B and 1C) and exceeding those achieved through Interleaved$_{T1,T2}$ training (Fig 3C and 3D).

We interpret these results as follows (see below for detailed synaptic connectivity analysis). Each episode of new Task 2 training improves Task 2 performance but damages synaptic connectivity responsible for old Task 1. If continuous Task 2 training is long enough, the damage to Task 1 becomes irreversible. Having a sleep phase after a short period of Task 2 training enables spontaneous forward replay between hidden and output layers (H->O) that preferentially benefits the strongest synapses. Thus, if Task 1 synapses are still strong enough to maintain replay, they are replayed and weights are increased.

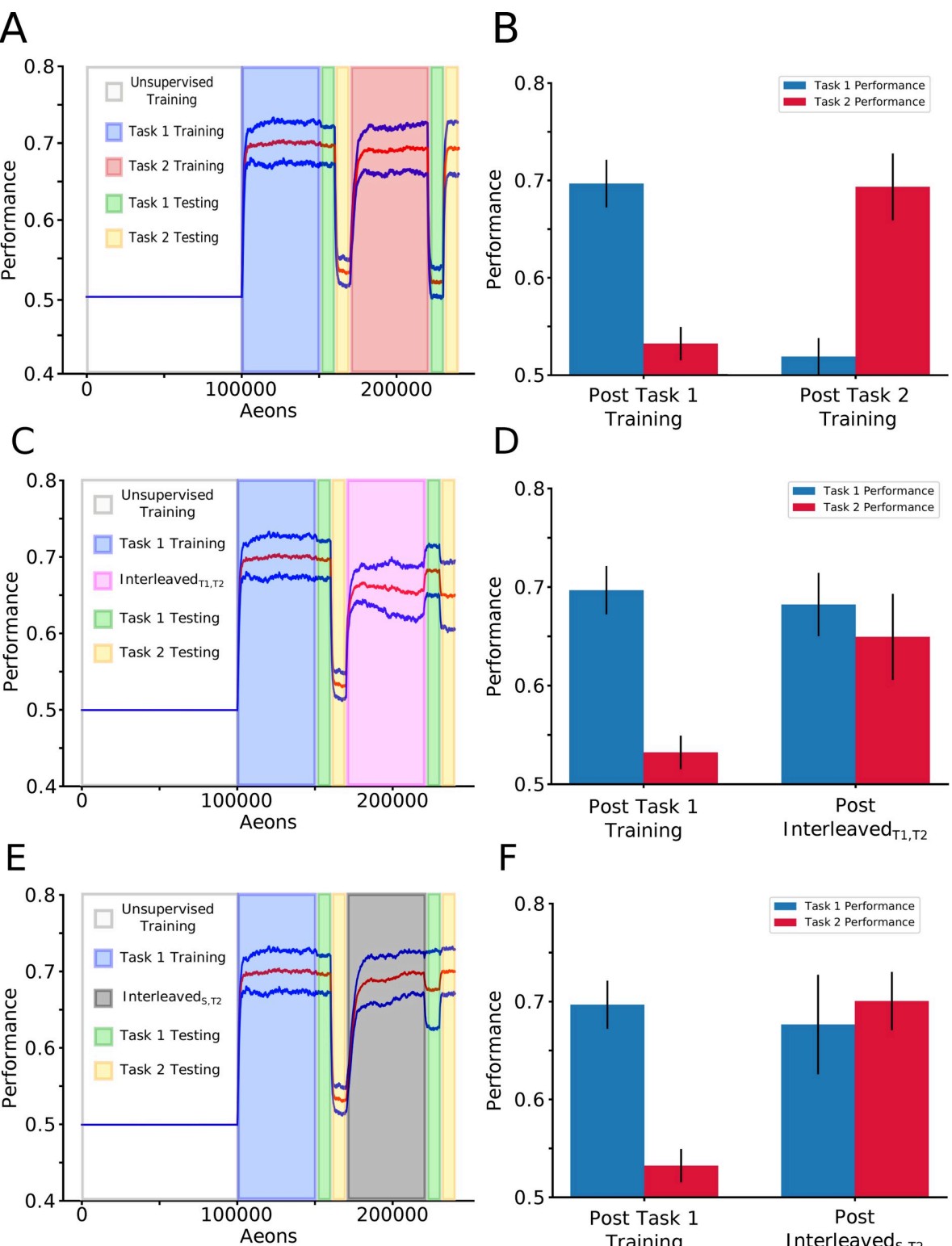

**Fig 3. Sleep prevents catastrophic forgetting during new task training.** **(A)** Mean performance (red line) and standard deviation (blue lines) over time: unsupervised training (white), Task 1 training (blue), Task 1/2 testing (green/yellow), Task 2 training (red), Task 1/2 testing (green/yellow). **(B)** Mean and standard deviation of performance during testing on Task 1 (blue) and Task 2 (red). Task 2 training after Task 1 training led to Task 1 forgetting. **(C)** Task paradigm similar to that shown in (A) but with Interleaved$_{T1,T2}$ training (pink) instead of Task 2 training. **(D)** Mean and standard deviation of performance during testing on Task 1 (blue) and Task 2 (red).

Interleaved$_{T1,T2}$ training allowed new Task 2 learning without forgetting old Task 1. **(E)** Task paradigm similar to that shown in (A) but with Interleaved$_{S,T2}$ training (gray) instead of Task 2 training. **(F)** Mean and standard deviation of performance during testing on Task 1 (blue) and Task 2 (red). Embedding sleep phases to the new Task 2 training protected old Task 1 memory.

## Sleep can protect synaptic configuration from previous training but does not provide training by itself

In simulations presented in Fig 3, during sleep phase, each hidden layer neuron was stimulated by noise, a Poisson distributed spike train, and we ensured that its firing rate during sleep would be close to the mean rate of that neuron firing across all the preceding training sessions. Therefore, intensity of the noise input during Interleaved$_{S,T2}$ was influenced by preceding Task 1 training and could also vary between H neurons. To eliminate the possibility that such input may provide direct Task 1 training during sleep, three additional experiments were conducted. First, we applied Interleaved$_{S,T1}$ phase to a completely naive network. Importantly, even though this network was never trained on Task 2, we used information about hidden layer neuron firing rates after Task 2 training from another experiment. In other words, we artificially took into account Task 2 firing rate data to design random input during sleep to check if this might be sufficient to improve the network performance on Task 2. We found that the network learns Task 1 but Task 2 performance remained at baseline (S4A and S4B Fig). In a second experiment, a similar period of Interleaved$_{S,T1}$ was applied following Task 1 training (S4C and S4D Fig) and we found that it maintained performance on Task 1 but again without any performance gain for Task 2.

In a third experiment, we repeated the sequence shown in Fig 3E, however, during the sleep phase, we provided each hidden layer neuron with a Poisson spike train input which was drawn (independently) from the same distribution, i.e., we used the same input firing rate for all hidden layer neurons determined by the mean firing of the entire hidden layer population as opposed to the private spiking history of individual H neurons in the Fig 3E and 3F experiments (termed Uniform-Noise Sleep (US)). The network's performance under this implementation of noise, Interleaved$_{US,T1}$, (S4E and S4F Fig) was similar to that from our original sleep implementation (see Fig 3E and 3F). Taken together, these results suggest that the properties of the input that drives firing during sleep are not essential to enable replay, any similar to awake random activity in layers H and O is sufficient to prevent forgetting.

## Sleep replay protects critical synapses of the old tasks

To reveal synaptic weights dynamics during training and sleep, we next traced "task-relevant" synapses, i.e. synapses identified in the top 10% of the distribution following training on that specific task. We first trained Task 1, followed by Task 2 training (Fig 4A) and we identified "task-relevant" synapses after each task training. Next, we continued by training Task 1 again but we interleaved it with periods of sleep: T1->T2->Interleaved$_{S,T1}$. Sequential training of Task 2 after Task 1 led to forgetting of Task 1, but after Interleaved$_{S,T1}$ Task 1 was relearned while Task 2 was preserved (Fig 4A and 4B), as in the experiments in the previous section (Fig 3C). Importantly, this protocol allowed us to compare synaptic weights after Interleaved$_{S,T1}$ training with those identified as task-relevant after individual Task 1 and Task 2 training (Fig 4C). The structure in the distribution of Task 1-relevant synapses formed following Task 1 training (Fig 4C; top-left) was destroyed following Task 2 training (top-middle) but partially recovered following Interleaved$_{S,T1}$ training (top-right). The distribution structure of Task 2-relevant synapses following Task 2 training (bottom-middle) was not present following Task 1 training (bottom-left) and was partially retained following Interleaved$_{S,T1}$ training (bottom-

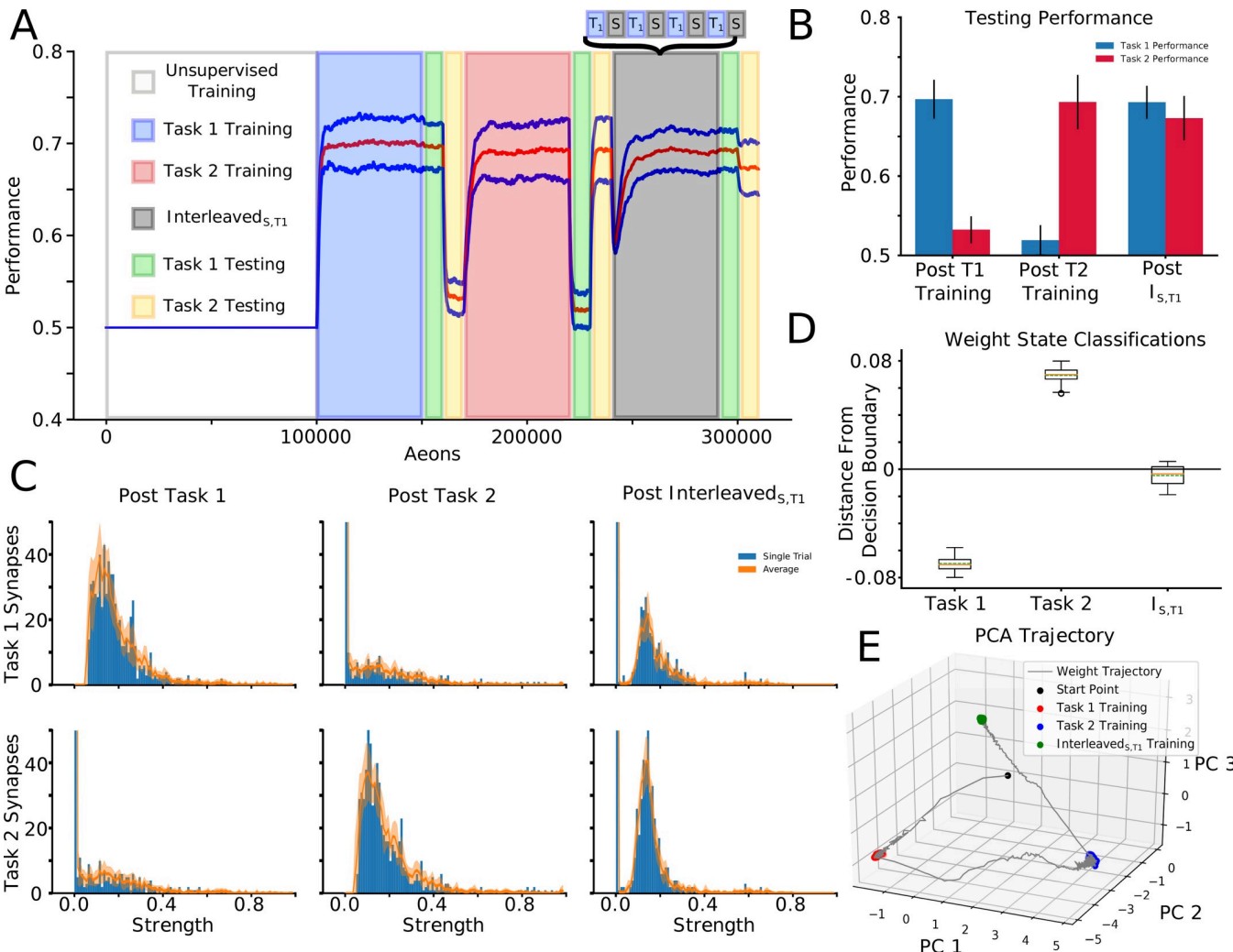

**Fig 4. Interleaving periods of new task training with sleep allows integrating synaptic information relevant to new task while preserving old task information. (A)** Mean performance (red line) and standard deviation (blue lines) over time: unsupervised training(white), Task 1 training (blue), Task 1/2 testing (green/yellow), Task 2 training (red), Task 1/2 testing (green/yellow), Interleaved$_{S,T1}$ training (grey), Task 1/2 testing (green/yellow). Note that performance for Task 2 remains high at the end despite no Task 2 training during Interleaved$_{S,T1}$. **(B)** Mean and standard deviation of performance during testing on Task 1 (blue) and Task 2 (red). **(C)** Distributions of task-relevant synaptic weights (blue bars–single trial, orange line / shaded region–mean / std across 10 trails). The distributional structure of Task 1-relevant synapses following Task 1 training (top-left) is destroyed following Task 2 training (top-middle), but partially recovered following Interleaved$_{S,T1}$ training (top-right). Similarly, the distributional structure of Task 2-relevant synapses following Task 2 training (bottom-middle), which was not present following Task 1 training (bottom-left), was partially preserved following Interleaved$_{S,T1}$ training (bottom-right). **(D)** Box plots with mean (dashed green line) and median (dashed orange line) of the distance to the decision boundary found by an SVM trained to classify Task 1 and Task 2 synaptic weight matrices for Task 1, Task 2, and Interleaved$_{S,T1}$ training across trials. Task 1 and Task 2 synaptic weight matrices had mean classification values of -0.069 and 0.069 respectively, while that of Interleaved$_{S,T1}$ training was -0.0047. **(E)** Trajectory of H to O layer synaptic weights through PC space. Synaptic weights which evolved during Interleaved$_{S,T1}$ training (green dots) clustered in a location of PC space intermediary between the clusters of synaptic weights which evolved during training on Task 1 (red dots) and Task 2 (blue dots).

right). It should be noted that this qualitative pattern can be distinctly observed in a single trial (Fig 4C; Blue Bars), but also generalizes across trials (Fig 4C; Orange Line). Thus, sleep can preserve important synapses while incorporating new ones.

To better understand the effect of Interleaved$_{S,T1}$ training on the synaptic weights, we trained a support vector machine (SVM; see *Method*: *Support Vector Machine Training* for details) to classify the synaptic weight configurations between layers H and O according to whether they serve to perform Task 1 or Task 2 on every trial. Fig 4D shows that the SVMs

robustly and consistently classified the synaptic weight states after Task 1 and Task 2 training while those after Interleaved$_{S,T1}$ fell significantly closer to the decision boundary. This indicates that the synaptic weight matrices which result from Interleaved$_{S,T1}$ training are a mixture of Task 1 and Task 2 states. Using principal components analysis (PCA), we found that while synaptic weight matrices associated with Task 1 and Task 2 training cluster in distinct regions of PC space, Interleaved$_{S,T1}$ training pushes the synaptic weights to an intermediate location between Task 1and Task 2 (Fig 4E). Importantly, the smoothness of this trajectory to its steady state suggests that Task 2 information is never completely erased during this evolution. We take this as evidence that Interleaved$_{S,T1}$ training is capable of integrating synaptic information relevant to Task 1 while protecting Task 2 information.

This analysis applied during interleaved training of Task 1 and Task 2 (Interleaved$_{T1,T2}$), revealed similar results (S5 Fig), suggesting that Interleaved$_{S,T1}$ can enable similar synaptic weights dynamics as Interleaved$_{T1,T2}$ training, but without access to the old task data (old training environment).

## Receptive fields of decision-making neurons after sleep represent multiple tasks

To confirm that the network had learned both tasks after Interleaved$_{S,T1}$ training, we visualized the receptive fields of decision-making neurons in layer O (Fig 5; see Fig 2 for comparison). Fig 5A shows the receptive field for the neuron in layer O which controls movement in the upper-left direction. This neuron responded to both horizontal (rewarded for Task 1) and vertical (rewarded for Task 2) orientations in the upper-left quadrant of the visual field. Although it initially appears that this layer O neuron may also be responsive to diagonal patterns in this region, analysis of the receptive fields of neurons in layer H (Fig 5B) revealed that these receptive fields are selective to either horizontal food particles (left six panels; rewarded for Task 1) or vertical food particles (right six panels; rewarded for Task 2) in the upper-left quadrant of the visual field. Other receptive fields were responsible for avoidance of punished particles for both tasks (see examples in Fig 5B, bottom-middle-right and bottom-middle-left). Thus, the network utilizes one of two distinct sets of layer H neurons, selective for either Task 1 or Task 2, depending on which food particles are present in the environment. To validate these qualitative results we inspected the PRM metrics for all food particle orientations across ten trials following Interleaved$_{S,T1}$ training. The comparatively high mean values for horizontal and vertical food particle orientations revealed the network's movement was significantly driven by these rewarded food particle orientations (horizontal and vertical), quantifying multitask memory integration into the network's synaptic weight matrix. (S3C Fig).

## Periods of sleep allow for integration of a new task memory without interference through renormalization of task-relevant synapses

To visualize synaptic weight dynamics during Interleaved$_{S,T1}$ training, traces of all synapses projecting to a single representative layer O neuron were plotted (Fig 6A). As in Fig 4, we wanted to monitor task specific synapses, so we used the training paradigm of T1->T2->Interleaved$_{S,T1}$, and Task 1 and Task 2 relevant synapses were identified after each individual task training. At the onset of Interleaved$_{S,T1}$ training (i.e. 240,000 aeons), the network was only able to perform on Task 2, meaning the strong synapses in the network were specific to this task. These synapses were represented by a cluster ranging from ~0.08 to ~0.4; the rest of synapses grouped near 0. As Interleaved$_{S,T1}$ training progressed, Task 1 specific synapses moved to the strong cluster and some, presumably less important, Task 2 synapses moved to the weak

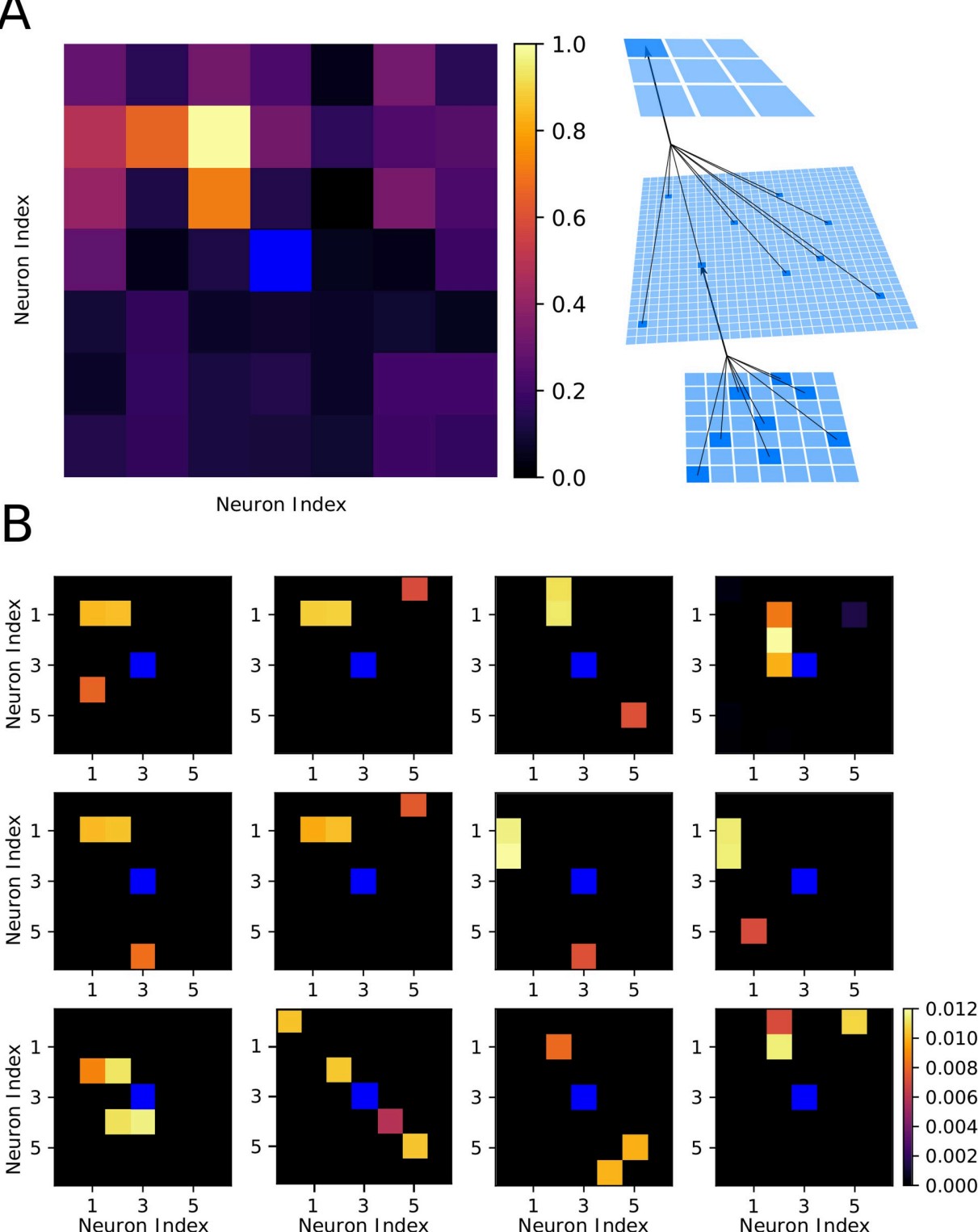

**Fig 5. Receptive fields following interleaved Sleep and Task 1 training reveal how the network can multiplex the complementary tasks. (A)**
Left, Receptive field of the output layer neuron controlling movement to the upper-left direction following interleaved sleep and Task 1 training.
This neuron has a complex receptive field capable of responding to horizontal and vertical orientations in the upper-left quadrant of the visual
field. Right, Schematic of the connectivity between layers. **(B)** Examples of receptive fields of hidden layer neurons which synapse strongly onto
the output neuron from (A) after interleaved Sleep and Task 1 training. The majority of these neurons selectively respond to horizontal food

particles (left half) or vertical food particles (right half) in the upper-left quadrant of the visual field, promoting movement in that direction and acquisition of the rewarded patters.

cluster. After a period of time the rate of transfer decreased and the total number of synapses in each group stabilized, showing that the network approached equilibrium (Fig 6B).

To visualize how sleep renormalizes task relevant synapses, we plotted two-dimensional weight distributions for T1->T2 (Fig 6C) and T2->Interleaved$_{S,T1}$ (Fig 6D) experiments (see *Methods*: *2-D Synaptic Weight Distributions* for details). To establish a baseline, in Fig 6C (left) the weight state at the end of Task 1 training (X-axis) (see overall timeline of this experiment in Fig 4A) was compared to itself (Y-axis). This formed a perfectly diagonal plot. The next comparison (Fig 6C, middle) was between the weight state after Task 1 training (X-axis) and a time early on Task 2 training (Y-axis). At that time, synapses were only able to modify their strength slightly, causing most points to lie close to the diagonal. As training on Task 2 continued, synapses moved far away from the diagonal (Fig 6C, right). Two trends were observed: (a) set of synapses that had a strength near zero following Task 1 training increased strength following Task 2 training (Fig 6D, right, red dots along Y-axis); (b) many strongly trained by Task 1 synapses were depressed down to zero (Fig 6C, right, red dots along X-axis). The latter illustrates the effect of catastrophic forgetting—complete overwriting of the synaptic weight matrix caused performance of Task 1 to return to baseline after training on Task 2.

Does sleep prevent overwriting of the synaptic weight matrix? Fig 6D plots used the weight state at the end of training Task 2 as a reference which is then compared to different times during Interleaved$_{S,T1}$ training. The first two plots (Fig 6D, left/middle) are similar to those in Fig 6C. However, after continuing Interleaved$_{S,T1}$ training (Fig 6D, right) many synapses that were strong following Task 2 training were not depressed to zero but rather were pushed to an intermediate strength (note cluster of points parallel to X-axis). Thus, Interleaved$_{S,T1}$ training preserved strong synapses from a previously learned task while also introducing new strong synapses to perform the new task.

Can we prevent old task forgetting simply by freezing a fraction of old task-relevant synapses to prevent their damage by new training? We found that freezing 1% of Task 1-relevant weights allowed Task 2 to be learned, but was not capable of preserving Task 1 (S6A Fig). Freezing 5% of Task 1-relevant weights (S6B Fig) resulted in modest performance on both tasks, but significantly below that seen after Interleaved$_{S,T2}$ (see Fig 3F). Finally, freezing 10% of Task 1-relevant weights (S6C Fig) was capable of fully preserving Task 1 performance, but prevented Task 2 from being learned.

Thus, in all cases, some degree of retroactive or prospective interference was observed highlighting the fact that the sleep-like phase performs a significantly more sophisticated modification to the weight matrix than simply freezing (or amplifying) task relevant synapses. Sleep is capable of intelligently selecting which certain strong synapses to maintain in addition to which weak synapses should be strengthened. Indeed, the sleep phase results in a large cluster of weights being renormalized around an intermediate value of synaptic strength in the network. This may also explain why we observed somewhat better overall performance (combined performance on both tasks) after sleep compare with interleaved training (see Fig 3). Indeed, interleaved training requires repetitive activation of the entire memory pattern, so if different memory patterns compete for synaptic resources then each phase of interleaved training will enhance one memory trace but damage the others. This is in contrast to spontaneous replay during sleep when only task specific subsets of neurons and synapses may be involved in each replay episode. It is worth mentioning that freezing a fraction of synaptic weights that are most relevant to old tasks (however, implemented in more complex form) is

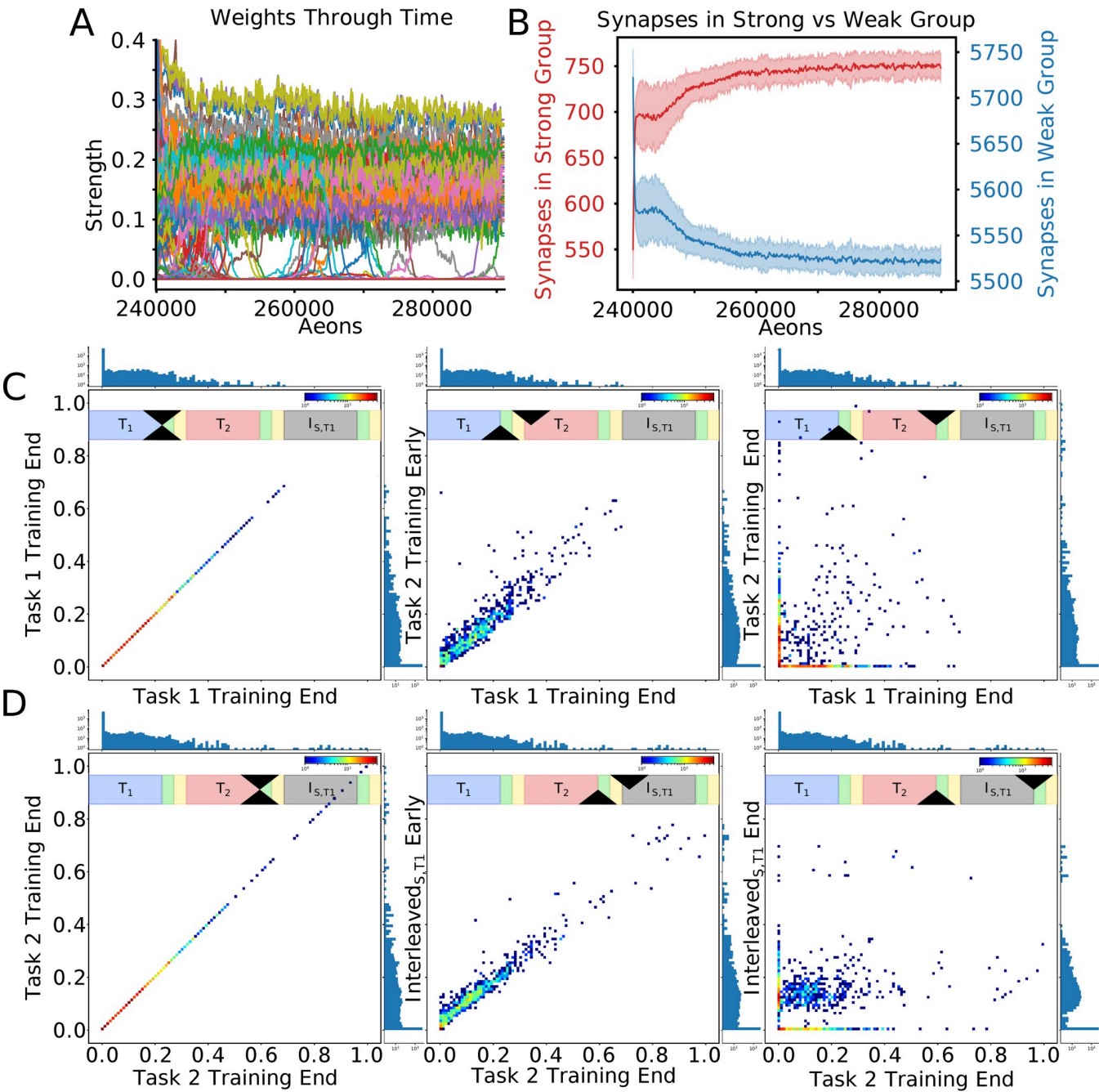

**Fig 6. Periods of sleep allow learning Task 1 without interference with old Task 2 through renormalization of task-relevant synapses.** (A) Dynamics of all incoming synapses to a single output layer neuron during Interleaved$_{S,T1}$ training shows the synapses separate into two clusters. The network was trained in the following order: T1->T2->Interleaved$_{S,T1}$. (B) Number of synapses in the strong (red) and weak (blue) clusters during Interleaved$_{S,T1}$. (C) Two-dimensional histograms illustrating synaptic weights dynamics. For each plot, the x-axis represents synaptic weight after Task 1 training and the y-axis represents the synaptic weight at a different point in time (Scale bar: brown—50 synapses/bin, blue—1 synapse/bin). One-dimensional projections along top and right sides show the global distribution of synapses at the time slices for a given plot. (D) Same as (C) except the x-axis refers to the end of Task 2 training. Note, that after a full period of Interleaved$_{S,T1}$ training (right), weak synapses were recruited to support Task 1 (red cluster along the y-axis) and many Task 2 specific synapses remained moderately strong (blue cluster along x-axis).

one of the approaches in machine learning to avoid catastrophic forgetting–Elastic Weight Consolidation [7].

## Periods of interleaved sleep and new task training push the network weight state towards the intersection of Task 1 and Task 2 synaptic weights configuration manifolds

Can many distinct synaptic weight configurations support a given task, or is each task supported by a unique synaptic connectivity matrix? Our previous analysis suggests that each task can be served by at least two different configurations–one unique for that task (Task 1 or Task 2) and another one that supports both Task 1 and Task 2. To further explore this question and to identify possible task-specific solution manifolds ($M_{T1}$ and $M_{T2}$) and their intersection ($M_{T1 \cap T2}$) in synaptic weights space, we used multiple trials of Task 1 and Task 2 training to sample the manifolds (Fig 7A). Here, red/blue dots indicate an exclusive high degree of performance on Task 1/2 respectively, while cyan and greed dots indicate states where the network is able to perform on both tasks simultaneously. Since this analysis was generated utilizing a wide variety of simulation paradigms with many corresponding trials differing in randomness, we believe it allows us to draw generalized conclusions. We therefore interpret these results as evidence that synaptic weight space includes a manifold, $M_{T1}$, where different configurations of weights (red, green, cyan dots) all allow for Task 1 to perform well. This manifold intersects with another one, $M_{T2}$, where different weights configurations (blue, green, cyan dots) are all suitable for Task 2. Fig 7B and 7C show 2D dimensionality reductions to PCA space, and include trajectories in addition to end states. One can see that PC 1 seems to capture the extent to which a synaptic weight configuration is associated with Task 1 (positive values) or Task 2 (negative values), while PC 2 and PC 3 capture the variance in synaptic weight configurations associated with Task 1 and Task 2, respectively. Note, the trajectories through this space (red/blue lines) during Interleaved$_{T1,T2}$ and Interleaved$_{S,T1/T2}$ training would also belong to the respective task manifolds as performance on the old tasks was never lost in these training scenarios.

We next calculated the distance from the current synaptic weight configurations to $M_{T1}$ (Fig 7D), $M_{T2}$ (Fig 7E), and $M_{T1 \cap T2}$ (Fig 7F; see *Methods*: *Distance from Solution Manifolds* for details) during different training protocols. Fig 7D and 7E show that while Sequential (T1->T2 or T2->T1) training causes synaptic weight configurations to diverge quickly from its initial solution manifold (i.e. $M_{T1}$ or $M_{T2}$) and to remain far (suggesting quick forgetting of the original task), both Interleaved$_{T1,T2}$ and Interleaved$_{S,T1/T2}$ training cause synaptic weight configurations to stay relatively close to the initial solution manifold as a new task was learned. (Note, that we certainly under sampled $M_{T1}$ and $M_{T2}$, which may explains initial distance increase.) Importantly, Fig 7F shows that both Interleaved$_{T1,T2}$ and Interleaved$_{S,T1/T2}$ training cause synaptic weight configurations to smoothly converge towards $M_{T1 \cap T2}$, while Sequential training avoids this intersection entirely.

In Fig 7G we present a schematic depiction of these results. The task-specific manifolds, $M_{T1}$ and $M_{T2}$, are depicted in 3D as two volumes whose boundaries are defined by two orthogonal elliptic paraboloids with opposite orientation. The ellipsoidal intersection approximates the volume comprising $M_{T1 \cap T2}$. Fig 7H and 7I depict a cartoon of trajectories taken by the network in this space following Task 2 and Task 1 training, respectively. Sequential training causes the network to jump directly from one task-specific solution manifold to the other, resulting in catastrophic forgetting. In contrast, interleaving new task training with sleep (Interleaved$_{S,T1/T2}$) prevents catastrophic forgetting by keeping the network close to the old task solution manifold as it converges towards $M_{T1 \cap T2}$ –a region capable of supporting both tasks simultaneously.

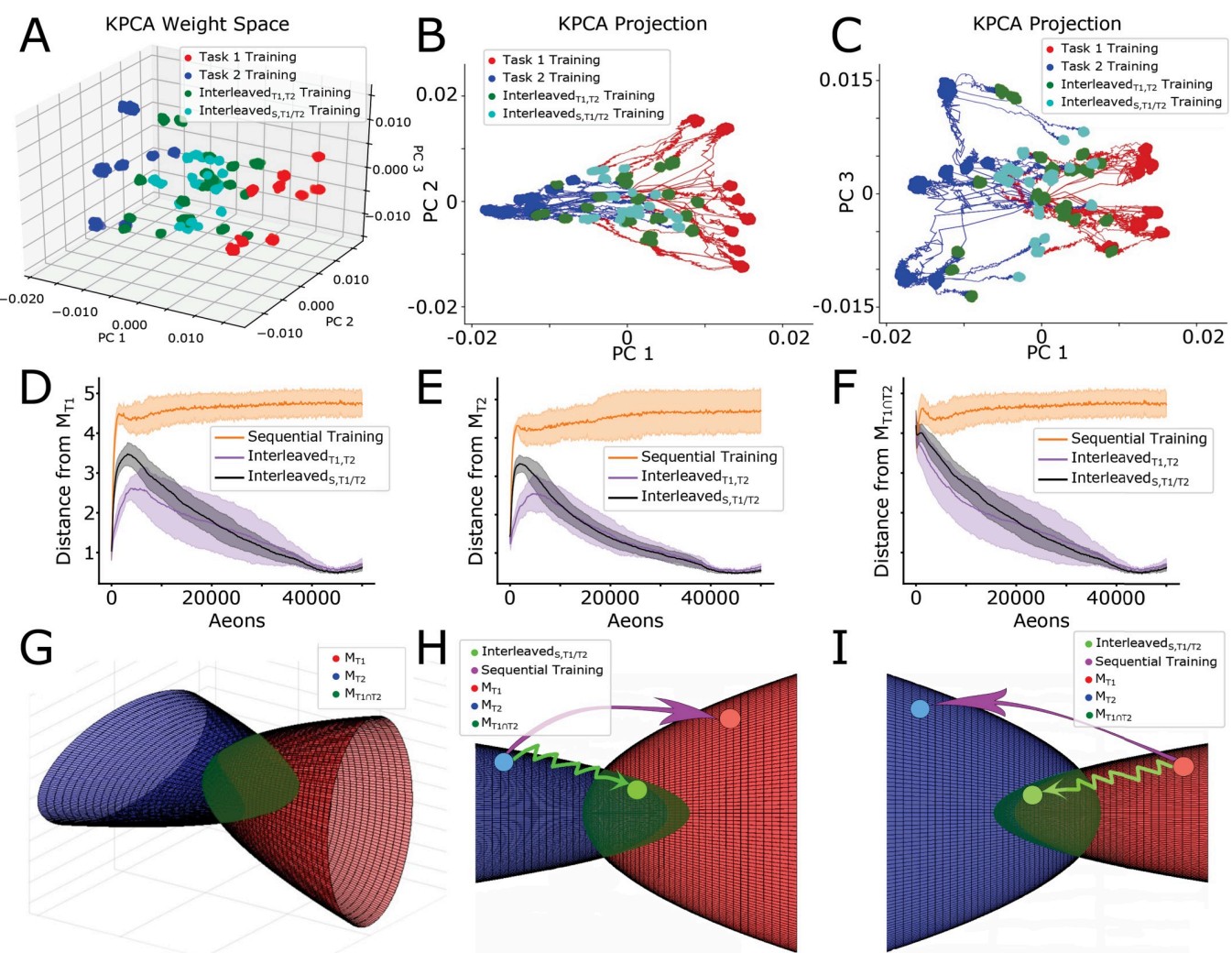

**Fig 7. Periods of sleep push the network towards the intersection of Task 1 and Task 2 synaptic weight manifolds. (A-C)** Low-dimensional visualizations of the synaptic weight configurations of 10 networks obtained through kPCA for 3-dimensions (A) and 2-dimensions (B-C). Synaptic weight configurations taken from the last fifth of Task 1 (red dots), Task 2 (blue dots), Interleaved$_{T1,T2}$ (green dots), and Interleaved$_{S,T1/T2}$ (cyan dots) training are shown. Trajectories resulting from Interleaved$_{T1,T2}$ and Interleaved$_{S,T1/T2}$ training following Task 1 (Task 2) training are shown in red (blue). **(D-F)** Average (solid lines) and standard deviation (shaded regions) of the Euclidean distances between the current synaptic weight configuration and $M_{T1}$ (D), $M_{T2}$ (E), and $M_{T1 \cap T2}$ (F) during Sequential (orange), Interleaved$_{T1,T2}$ (purple), and Interleaved$_{S,T1/T2}$ (black) training. **(G)** Cartoon illustration of the task-specific point-sets shown in (A-C) as solution manifolds $M_{T1}$ (red) and $M_{T2}$ (blue). $M_{T1}$ and $M_{T2}$ can be thought of as two volumes with boundaries defined by the interiors of oppositely oriented elliptic paraboloids which intersect orthogonally defining an approximately ellipsoidal volume near the origin ($M_{T1 \cap T2}$; dark green). **(H, I)** Sequential training (pink arrow) causes the network to jump from one solution manifold to the other while avoiding $M_{T1 \cap T2}$, while Interleaved$_{S,T1/T2}$ training (light green arrow) keep the network close to the initial solution manifold as it converges towards $M_{T1 \cap T2}$.

## Discussion

We report that a multi-layer spiking neural network utilizing reinforcement learning exhibits catastrophic forgetting upon sequential training of two complementary complex foraging tasks, however the problem is mitigated if the network is allowed, during new task training, to undergo intervening periods of spontaneous reactivation which are equivalent to the periods of sleep in a biological brain. Old task was spontaneously replayed during sleep, therefore interleaving new task training with sleep was effectively equivalent to explicit interleaved training of the old and new tasks without the need to store and train on previous task data or environments. At the synaptic level, training a new task alone led to complete overwriting of

synaptic weights responsible for the old task. In contrast, interleaving periods of reinforcement learning on a new task with periods of unsupervised plasticity during sleep preserved critical old task synapses to avoid forgetting and enhanced synapses relevant for a new task to allow new task learning. Thus, in synaptic weight space, the network weight configuration was pushed towards the intersection of the manifolds representing synaptic weight configurations associated with individual tasks—an optimal compromise for performing both tasks.

The critical role that sleep plays in learning and memory is supported by a vast, interdisciplinary literature spanning both psychology and neuroscience [16,22,43–45]. Specifically, it has been suggested that REM sleep supports the consolidation of non-declarative or procedural memories while non-REM sleep supports the consolidation of declarative memories [16,21,22]. In particular, REM sleep has been shown to be important for the consolidation of memories of hippocampus-independent tasks involving perceptual pattern separation, such as the texture discrimination task [16,46]. Despite the difference in the cellular and network dynamics during these two stages of sleep [16,22], both are thought to contribute to memory consolidation through repeated reactivation, or replay, of specific memory traces acquired during learning [16,21,39,44,47–49]. These studies suggest that through replay, sleep can support the process of off-line memory consolidation to circumvent the problem of catastrophic forgetting.

From mechanistic perspective, the sleep phase in our model protects old memories by enabling spontaneous reactivation of neurons and changing synapses responsible for previously learned tasks. We previously reported that in the thalamocortical model a sleep phase may enable replay of spike sequences learned in awake to improve post-sleep performance [38–40] and to protect old memories from catastrophic forgetting [41]. Here we found, however, that a single episode of new task training using reinforcement learning could quickly erase old memories to the point that they cannot be recovered by subsequent sleep. The solution was similar to how the brain slowly learns procedural (hippocampal-independent) memories [16,21,22,46,50]. Each episode of new task training improves new task performance only slightly but also damages slightly synaptic connectivity responsible for the older task. Subsequent sleep phases enable replay that preferentially benefits the strongest synapses, such as those from old memory traces, to allow them to recover.

We found that multiple distinct configurations of synaptic weights can support each task, suggesting the existence of task specific solution manifolds in synaptic weight space. Sequential training of new tasks makes the network to jump from one solution manifold to another, enabling memory for the most recent task but erasing memories of the previous tasks. Interleaving new task training with sleep phases enables the system to evolve towards intersection of these manifolds where synaptic weight configurations can support multiple tasks (a similar idea was recently proposed in the machine learning literature to minimize catastrophic interference by learning representations that accelerate future learning [51]). From this point of view having multiple episodes of new task training interleaved with multiple sleep episodes allows gradual convergence to the intersection of the manifolds representing old and new tasks, while staying close to the old task manifold. In contrast, a single long episode of new task learning would push the network far away from the old task manifold making it impossible to recover by subsequent sleep.

Although classical interleaved training of the old and new tasks showed similar performance results in our model as interleaving new task training with sleep, we believe the latter to be superior on the following theoretical grounds. Classical interleaved training will necessarily cause the system to oscillate about the optimal location in synaptic weight space which can support both tasks because each training cycle uses a cost function specific to only a single task. While this can be ameliorated with a learning rate decay schedule, the system is never actually optimizing for the desired dual-task state. Sleep, on the other hand, can support not

only replays of the old task, but also support replays which are a mixture of both tasks [41,52,53]. Thus, through unsupervised plasticity during sleep replay, the system is able to perform approximate optimization for the desired dual-task (or multi-task) state.

Our results are in line with a large body of literature suggesting that interleaved training is capable of mitigating catastrophic forgetting in ANNs [4,10,11] and SNNs [12,13], which led to a number of replay-like algorithms involving storing a subset of previous veridical inputs and mixing them with more recent inputs to update the networks (reviewed in [9]). The novel contribution from our study is that the data intensive process of storing old data and using them for retraining can be avoided in SNN by implementing periods of noise-induced spontaneous reactivation during new task training; similar to how brains undergo offline consolidation periods during sleep resulting in reduced retroactive interference to previously learned tasks [16,50]. Indeed, we recently successfully implemented a similar approach in feedforward ANNs, where sleep-like phase prevented catastrophic forgetting and improved generalization and adversarial robustness [54–56]. And our results are in line with previous work done in humans showing that perceptual learning tasks are subject to retroactive interference by competing memories without an intervening period of REM sleep [21,46]. Moreover, performance on visual discrimination tasks in particular have been shown to steadily improve over successive nights of sleep [46], consistent with our findings that interleaving multiple periods of sleep with novel task learning leads to optimal performance on each task.

In comparing our modeling results to those found in the literature on biological learning, it is important to note an important difference in the "baseline" state of an animal undergoing an experimental training condition versus a neural network model. In our model, and indeed in all neural network models, the system begins as a "blank slate" without knowledge of any previous learning or competing demands. In contrast, animals under experimental training paradigms have a wealth of experiences which would serve as priors to bias the subsequent learning during training, leading potentially to proactive interference. Moreover, training is typically conducted across multiple days, with intervening periods during which the animal will be subject to an array of various task-irrelevant stimuli and organismal demands possibly leading to retroactive interference. Both of these ensure that the baseline state of the animal entering a given training session is far from that of the "blank slate" a neural network model enters with, as well as that recently learned memories may start degrading quickly in the brain while the network weights remain unchanged post training (unless new task is explicitly trained). Due to this stark differences, we focus our attention on the interference phenomena which follow training on an initial task as opposed to initial learning. Viewed from this perspective, initial task training in our network can serve a similar role to the prior personal history of an animal subject.

While our model represents a dramatic simplification of a living system, we believe that it captures some important elements of how animal and human brains interact with the external world. The primary visual system is believed to employ a sequence of processing steps when visual information is increasingly represented by neurons encoding higher level features [35–37]. In insects, complex patterns of olfactory receptors activation by odors are encoded by sparse patterns of the mushroom body Kenyon cells firing [57–59]. This processing step is also similar to the function performed by convolutional layers of an ANN [34] and it was reduced to very simple convolution from the input to hidden layer in our model. Subsequently, in the vertebrate brain, associative areas and motor cortex are trained to make decisions based on reward signals released by neuro modulatory centers [10,60–62]. In insects, Kenyon cells make plastic (subject to rewarded STDP) projections to the lobes [27,63]. This was reduced in our model to synaptic projections from the hidden to output (decision making) layer implementing rewarded STDP to learn a task [30–32]. While NREM sleep in vertebrates is characterized

by complex patterns of synchronized neuronal activity [16], REM sleep is characterized by low-synchronized firing [42], similar to activity during sleep-like phase in our model and paradoxical sleep with similar properties has been reported in honeybee and fruit fly [64–66].

Our study predicts synaptic level mechanisms of how sleep-based memory reactivation can protect old memory traces during training of a new interfering memory task. It suggests that, at least for procedural memories that are directly encoded to the cortical network connectivity during new training, multiple episodes of training interleaved with periods of sleep provide necessary mechanisms to prevent forgetting old memories. Interleaving new task training with sleep enables the connectivity matrix to evolve towards the joint synaptic weight configuration, representing the intersection of manifolds supporting individual tasks. Sleep makes this possible by replaying old memory traces without explicit usage of the old training data.

## Methods

### Environment

Foraging behavior took place in a virtual environment consisting of a 50x50 grid with randomly distributed "food" particles. Each particle was two pixels in length and could be classified into one of four types depending on its orientation: vertical, horizontal, positively sloped diagonal, or negatively sloped diagonal. During the initial unsupervised training period, the particles are distributed at random with the constraints that each of the four types are equally represented and no two particles can be directly adjacent. During training and testing periods only the task-relevant particles were present. When a particle was acquired as a result of the virtual agent moving, it was removed from its current location (simulating consumption) and randomly assigned to a new location on the grid, again with the constraint that it not be directly adjacent to another particle. This ensures a continuously changing environment with a constant particle density. The density of particles in the environment was set to 10%. The virtual agent can see a 7x7 grid of squares (the "visual field") centered on its current location and it could move to any adjacent square, including diagonally, for a total of eight directions.

### Network structure

The network was composed of 842 spiking reduced (map-based) model neurons (see *Methods*: *Map-based neuron model* below) [67,68], arranged into three feed-forward layers to mimic a basic biological circuit: a 7x7 input layer (I), a 28x28 hidden layer (H), and a 3x3 output layer (O) with a nonfunctional center neuron (Fig 1). Input to the network was simulated as a set of suprathreshold inputs to the neurons in layer I, equivalent to the lower levels of the visual system, which represent the position of particles in an egocentric reference frame relative to the virtual agent (positioned in the center of the 7x7 visual field). The most active neuron in layer O, playing the role of biological motor cortex, determined the direction of the subsequent movement. Each neuron in layer H, which can be loosely defined as higher levels of the visual system or associative cortex, received excitatory synapses from 9 randomly selected neurons inlayer I. These connections initially had random strengths drawn from a normal distribution. Each neuron in layer H connected to every neuron in layer O with both an excitatory ($Wij$) and an inhibitory ($WIij$) synapse. This provided an all-to-all connectivity pattern between these two layers and accomplished a balanced feed-forward inhibition [69] found in many biological structures [69–74]. Initially, all these connections had uniform strengths and the responses in layer O were due to the random synaptic variability. Random variability was a property of all synaptic interactions between neurons and was implemented as variability in the magnitude of the individual synaptic events.

## Policy

Simulation time was divided up into epochs of 600 timesteps, each roughly equivalent to 300 ms. At the start of each epoch the virtual agent received input corresponding to locations of nearby particles within the 7x7 "visual field". Thus 48 of the 49 neurons in layer I received input from a unique location relative to the virtual agent. At the end of the epoch the virtual agent made a single move based on the activity in layer O. If the virtual agent moved to a grid location with a "food" particle present, the particle was removed and assigned to a randomly selected new location.

Each epoch was of sufficient duration for the network to receive inputs, propagate activity forward, produce outputs, and return to a resting state. Neurons in layer I which represent locations in the visual field containing particles received a brief pulse of excitatory stimulation sufficient to trigger a spike; this stimulation was applied at the start of each movement cycle (epoch). At the end of each epoch the virtual agent moved according to the activity which has occurred in layer O. Each simulation consisted of millions of these movement cycles / epochs, therefore a unit of time was introduced termed aeon (1 aeon = 100 epochs) for concise reporting.

The activity in layer O controlled the direction of the virtual agent's movement. Each of the neurons in layer O mapped onto a specific direction (i.e. one of the eight adjacent locations or the current location). The neuron in layer O which spiked the greatest number of times during the first half of the epoch defined the direction of movement for that epoch. If there was a tie, the direction was chosen at random from the set of tied directions. If no neurons in layer O spiked, the virtual agent continued in the direction it had moved during the previous epoch.

There was a 1% chance on every move that the virtual agent would ignore the activity inlayer O and instead move in a random direction. Moreover, for every movement cycle that passed without the virtual agent acquiring a particle, this probability was increased by 1%. The random variability promoted exploration vs exploitation dynamics and essentially prevented the virtual agent from getting stuck in movement patterns corresponding to infinite loops. While biological systems could utilize various different mechanisms to achieve the same goal, the method we implemented was efficient and effective for the scope of our study.

## Neuron models

For all neurons we used spiking model identical to the model used in in [14,15] that can be described by the following set of difference equations [68,75,76]:

$$V_{n+1} = f_\alpha(V_n, I_n + \beta_n),$$

$$I_{n+1} = I_n - \mu(V_n + 1) + \mu\sigma + \mu\sigma_n,$$

where $Vn$ is the membrane potential, $In$ is a slow dynamical variable describing the effects of slow conductances, and $n$ is a discrete time-step (0.5 ms). Slow temporal evolution of $In$ was achieved by using small values of the parameter $\mu << 1$. Input variables $\beta_n$ and $\sigma_n$ were used to incorporate external current $I^{ext}n$ (e.g. background synaptic input): $\beta_n = \beta^e I^{ext}{}_n$, $\sigma_n = \sigma^e I^{ext}{}_n$. Parameter values were set to $\sigma = 0.06$, $\beta^e = 0.133$, $\sigma^e = 1$, and $\mu = 0.0005$. The nonlinearity $f\alpha$ $(V_n, I_n)$ was defined in the form of the piece-wise continuous function:

$$f_\alpha(V_n, I_n) = \begin{cases} \alpha(1 - V_n)^{-1} + I_n, & V_n \leq 0 \\ \alpha + I_n, & 0 < V_n < \alpha + I_n \,\&\, V_{n-1} \leq 0 \\ -1 & \alpha + I_n \leq V_n \; or \; V_{n-1} > 0, \end{cases}$$

where $\alpha = 3.65$. This model is very computationally efficient, and, despite its intrinsic low dimensionality, produces a rich repertoire of dynamics capable of mimicking the dynamics of Hodgkin-Huxley type neurons both at the single neuron level and in the context of network dynamics [68,75,77].

To model the synaptic interactions, we used the following piece-wise difference equation:

$$g_{n+1}^{syn} = \gamma g_n^{syn} + \begin{cases} (1 - R + 2XR)g_{syn}/W_j, & spike_{pre} \\ 0, & \text{otherwise,} \end{cases}$$

$$I_n^{syn} = -g_n^{syn}(V_n^{post} - V_{rp}).$$

Here $gsyn$ is the strength of the synaptic coupling, modulated by the target rate $Wj$ of receiving neuron $j$. Indices $pre$ and $post$ stand for the pre- and post-synaptic variables, respectively. The first condition, $spikepre$, is satisfied when the pre-synaptic spikes are generated. Parameter $\gamma$ controls the relaxation rate of synaptic current after a presynaptic spike is received ($0 \leq \gamma < 1$). The parameter $R$ is the coefficient of variability in synaptic release. The standard value of $R$ is 0.12. $X$ is a random variable sampled from a uniform distribution with range [0, 1]. Parameter $Vrp$ defines the reversal potential and, therefore, the type of synapse (i.e. excitatory or inhibitory). The term $(1-R+2XR)$ introduces a variability in synaptic release such that the effect of any synaptic interaction has an amplitude that is pulled from a uniform distribution with range [1-R,1+R] multiplied by the average value of the synapse.

## Synaptic plasticity

Synaptic plasticity closely followed the rules introduced in [14,15]. A rewarded STDP rule [30–33] was operated on synapses between layers H and O while a standard STDP rule operated on synapses between layers I and H. A spike in a post-synaptic neuron that directly followed a spike in a pre-synaptic neuron created a *pre before post* event while the converse created a *post before pre* event. Each new post-synaptic (pre-synaptic) spike was compared to all pre-synaptic (post-synaptic) spikes with a time window of 120 iterations.

The value of an STDP event (trace) was calculated using the following equation [28,29]:

$$p = \frac{-|t_r - t_p|}{T_c},$$

$$tr_k = Ke^p$$

where $t_r$ and $t_p$ are the times at which the pre- and post-synaptic spike events occurred respectively, $Tc$ is the time constant and is set to 40 ms, and $K$ is maximum value of the trace $tr_k$ and is set to -0.04 for a *post before pre* event and 0.04 for a *pre before post* event.

A trace was immediately applied to synapse between neurons in layers I and H. However, for synapses between neurons in layers H and O the traces were stored for 6 epochs after its creation before being erased. During storage, a trace had an effect whenever there was a rewarding or punishing event. In such a case, the synaptic weights are updated as follows:

$$W_{ij} \leftarrow W_{ij} \prod_{k}^{traces} \left(1 + \frac{W_{i0}}{W_i} * \Delta_k\right),$$

$$\Delta_k = S_{rp}\left(\frac{tr_k}{t - t_k + c}\right)\frac{Sum_{tr}}{Avg_{tr}},$$

$$Sum_{tr} = \sum_{k}^{traces} \frac{tr_k}{t - t_k + c},$$

$$Avg_{tr} \leftarrow (1 - \delta)Avg_{tr} + \delta Sum_{tr},$$

where $t$ is the current timestep, $S_{rp}$ is a scaling factor for reward/punishment, $trk$ is the magnitude of the trace, $tk$ is the time of the trace event, $c$ is a constant (= 1 epoch) used for decreasing sensitivity to very recent spikes, $W_i = \Sigma_j W_{ij}$ is the total synaptic strength of all connections from the neuron $i$ in layer H to all neurons in layer O, $W_{i0}$ is a constant that is set to the initial value(*target value*) of $Wi$ at the beginning of the simulation. The term $W_{i0}/W_i$ helped to keep the output weight sum close to the initial target value. The effect of these rules was that neurons with lower total output strength could increase their output strength more easily.

The network was rewarded when the virtual agent moved to a location which contained a particle from a "food" pattern (horizontal in Task 1, vertical in Task 2) and $S_{rp} = 1$, and received a punishment of $S_{rp}$ = -0.001 when it moved to a location with a particle from a neutral pattern(negative/positive diagonal in Task 1/2). A small punishment of $S_{rp}$ = -0.0001 was applied if the agent moved to a location without a particle present to help the virtual agent learn to acquire "food" as rapidly as possible. During periods of sleep the network received a constant reward of $S_{rp}$ = 0.5 on each movement cycle.

To ensure that neurons in layer O maintained a relatively constant long-term firing rate, the model incorporated homeostatic synaptic scaling which was applied every epoch. Each timestep, the total strength of synaptic inputs $W_j = \Sigma_i W_{ij}$ to a given neuron in layer O was set equal to the target synaptic input $W_{j0}$ –a slow variable which varied over many epochs depending on the activity of the given neuron in layer O–which was updated according to:

$$W_{j0} \leftarrow \begin{cases} W_{j0}(1 + D_{tar}) \text{ spike rate} < \text{target rate} \\ W_{j0}(1 - D_{tar}) \text{ spike rate} > \text{target rate} \end{cases}$$

To ensure that the net synaptic input $W_j$ to any neuron was unaffected by plasticity events at the individual synapses at distinct timesteps and equal to $W_{j0}$, we implemented a scaling process akin to heterosynaptic plasticity which occurs after each STDP event. When any excitatory synapse of neuron in layer O changed in strength, all other excitatory synapses received by that neuron were updated according to:

$$W_{ij} \leftarrow W_{ij} \frac{W_{j0}}{\sum_i W_{ij}}$$

Additionally, all inhibitory synapses were modified via a similar heterosynaptic update rule following each STDP event where the strength of every outgoing inhibitory weight from a given neuron was set to the negative mean of all outgoing excitatory synapses of that same neuron. More rigorously:

$$WI_{ij} \leftarrow -\frac{1}{|j|} \sum_j W_{ij}$$

## Simulated sleep

To simulate the sleep phase, we inactive the sensory receptors (i.e. the input layer of network), cut off all sensory signals (i.e. remove all particles from the environment), and decouple output

layer activity from motor control (i.e. the output layer can spike but no longer causes the agent to move). We also change the learning rule between the hidden and output layer from rewarded to unsupervised STDP (see *Methods*: *Synaptic Plasticity* for details) as there is no way to evaluate decision-making without sensory input or motor output.

To simulate the spontaneous activity observed during REM sleep, we provided noise to each neuron in the hidden layer in a way which ensured that the spiking statistics of each neuron was conserved across awake and sleep phases. To determine these spiking rates, we recorded average spiking rates of neurons in the hidden layer H during preceding training of both Task 1 and Task 2; these task specific spiking rates were then averaged to generate target spiking rates for hidden layer neurons. Interleaved$_{S,T1}$ training consisted of alternating intervals of this sleep phase and training on Task 1, with each interval lasting 100 movement cycles (although no movement occurred).

## Support vector machine training

A support vector machine with a radial basis function kernel was trained to classify synaptic weight configurations as being related to Task 1 or Task2. Labeled training data were obtained by taking the excitatory synaptic weight matrices between the hidden and output layers from the last fifth of the Task 1 and Task 2 training phases (i.e. after performance had appeared to asymptote). These synaptic weight matrices were then flattened into column vectors, and the column vectors were concatenated to form a training data matrix of size *number of features* x *number of samples*. The number of features was equal to the total number of excitatory synapses between the hidden and output layer– 6272 dimensions. We then used this support vector machine to classify held out synaptic weight configurations from Task 1 and Task 2 training, as well as ones which resulted from Interleaved$_{T1,T2}$ and Interleaved$_{S,T1}$ training.

## 2-D synaptic weight distributions (Fig 6)

First for each synapse we found how its synaptic strength changes between two slices in time, where the given synapse's strength at time slice 1 is the point's X-value and strength at time slice 2 is its Y-value. Then we binned this space and counted synapses in each bin to make two dimensional histograms where blue color corresponds to a single synapse found in a bin and brown corresponds to the max of 50 synapses. These two-dimensional histograms assist in visualizing the movement of all synapses between the two slices in time that are specified by the timelines at the top of each plot. Conceptually, it is important to note that if a synapse does not change in strength between time slice 1 and time slice 2, then point the synapse corresponds to in this space will lie on the diagonal of the plot since the X-value will match the Y-value. If a great change in the synapse's strength has occurred between time slice 1 and time slice 2, then the synapse's corresponding point will lie far from the diagonal since the X-value will be distant from the Y-value. The points on the X-(Y-) axis represent synapses that lost (gained) all synaptic strength between time slice 1 and time slice 2.

## Distance from solution manifolds (Fig 7)

Each of the two solution manifolds (i.e. Task 1 and Task 2 specific manifolds) were defined by the point-sets in synaptic weight space which were capable of supporting robust performance on that particular task, namely the sets $M_{T1}$ and $M_{T2}$. This included the synaptic weight states from the last fifth of training on a particular task(i.e. after performance on that task appeared to asymptote) and all of the synaptic weight states from the last fifth of both Interleaved$_{T1,T2}$ and Interleaved$_{S,T1/T2}$ training. The intersection of the two solution manifolds (i.e. the point-set $M_{T1\cap T2}$) was defined solely by the synaptic weight states from the last fifth of both

Interleaved$_{T1,T2}$ and Interleaved$_{S,T1}$ training. As the network evolved along its trajectory in synaptic weight space, the distance from the current point in synaptic weight space, $pt$, to the two solution manifolds and their intersection were computed as follows:

$$d^n(p_t, M_\tau) = \min_{x \in M\tau}(d^n(p_t, x)).$$

Here, $d^n$ is the n-dimensional Euclidean-distance function, where $n$ is the dimensionality of synaptic weight space (i.e. $n = 6272$ here), $M_\tau$ is the point-set specific to the manifold or intersection in question (i.e. either $M_{T1}$, $M_{T2}$, or $M_{T1 \cap T2}$), and $x$ is a particular element of the point-set $M_\tau$.

## Particle responsiveness metric (PRM)

The particle responsiveness metric (PRM) developed to quantify how responsive the network's weight matrix is to specific food particle orientations thereby allowing the quality of the receptive field for a given task to be determined was defined as follows:

$$\text{PRM(Particle Type)} =$$
$$\sum_{\forall O \in Output} \text{grand(DirectionMask(O)} \odot \sum_{\forall H \in Hidden} W_{H \to O} * \sum_{\forall P \in Particle Masks} (W_H \odot P) * \text{grand}(W_H \odot P)^2)$$

Here *Output* is the set of all output layer neurons, $O$; *Hidden* is the set of all hidden layer neurons, $H$; *ParticleMasks* is the set of masks, $P$, representing all possible locations of a single instance of a *ParticleType* in the input field (e.g., horizontal bars would be a set of masks with a single horizontal bar placed in all possible locations in the visual field; each particle mask $P$ consists of a *7 x 7* matrix of zeros with ones being placed in locations that correspond to current food pixels). $W_H$ is a *7x7* synaptic weights matrix of a given hidden layer neuron $H$; $\odot$ gives Hadamard (or element-wise) product of two matrixes, $grand(A)$ is a grand sum of all the elements of a matrix A *($grand(A) = e^T A e$, where $e$ is all-ones vector)*. *DirectionMask(O)* takes in an output layer neuron, $O$, and returns a matrix that represents the direction of motion with respect to the input field. For example, when the neuron that directs the critter to move up and to the left is supplied as input, the function returns a *7 x 7* matrix of zeros with the top left *3 x 3* submatrix being ones. $W_{H \to O}$ simply returns the synapse strength from the source *(H)* to destination *(O)* neuron.

Although this is seemingly an intricate metric, it captures many desired features of the network's connectivity and responses to food particles present in the visual field. Conceptually, this metric is similar to the method used for developing the receptive fields of output layer neurons with respect to the input field (Figs 2 and 5). PRM builds upon this qualitative visualization, allowing us to numerically assess how specific particles influence output layer neurons to spike when present in the portion of the visual field that corresponds to the direction of motion for that neuron. The intuitions of the metric are as follows: $W_H \odot P$ develops a notion of how well the current hidden neuron's ($H$) connections to the input layer overlaps with the current food particle ($P$) placed at specific location. The resulting matrix is then multiplied by $grand(W_H \odot P)^2$, which emphasizes contribution of the $H$ neurons receiving input from adjacent pixels in correct orientation (i.e., sensitive to the food particles) vs those receiving input from random pixels. Indeed, when a hidden layer neuron $H$ overlaps strongly with a food particle $P$, the chances of spiking are significantly increased, thus this nonlinear term captures the high impact overlapping receptive fields and food particles has on output layer activity. $W_{H \to O}$ captures how strongly the current output layer neuron $O$ is listening to the current hidden layer neuron $H$.

These described pieces are multiplied together to form a weighted input receptive field of the output layer neuron with respect to a specific hidden layer neuron and food particle type / location. The sum of these terms for all hidden layer neurons and food particle locations is taken for a single output layer neuron, achieving a global view of all hidden layer neurons and food particle types / locations influencing the current output layer neuron. The *grand*(*A*) operation between the *DirectionMask*(*O*) and the previously described summed term is then taken to see how much the summed weighted receptive fields overlap with the corresponding direction of movement for output neuron *O*. This process is repeated for all output layer neurons to get a global quantification of how the current food particle influences activity in the direction of motion for all output layer neurons. When this metric is calculated for a given network state across food particle types we can observe what food particles impact output layer activity and drive the critter to move, highlighting what particle orientations the network is attracted to.

## Supporting information

**S1 Fig. Spike rasters showing network activity across various training regimes. (A-D)** Representative spike rasters from various training regimes. The vertical axis specifies a unique neuron in the network while time in epochs is shown horizontally. Here a single dot represents a specific neuron spiking at a given time while the color of the dot dictates what layer that neuron belongs to (green, blue, red corresponding to input, hidden, and output layers respectively). Panels A, B, C, D correspond to sample activity from Task 1 training, Task 2 training, $I_{T1,T2}$ training and $I_{S,T1}$ training respectively. Note, in panel D activity is taken during a period of sleep when the hidden layer is spontaneously activated. Thus, there are hidden (blue) and output (red) layer spikes while the input (green) layer is completely silent.
(EPS)

**S2 Fig. Model displays graceful degradation in performance as a result of hidden layer dropout. (A)** Mean performance (red line) and standard deviation (blue lines) over time: unsupervised training (white), Task 1 training (blue), Task 1 testing (green). Hidden layer neurons are randomly removed during testing period. Gradient bar above Task 1 testing (green) displays the number of hidden layer neurons over time starting at 784 and decreasing down to 0. The testing performance remains high until ~25% of neurons are left, after which it starts to drop. This highlights the formation of a distributed synaptic structure between hidden and output layer neurons developed during training, ensuring output layer activity is not dictated by a select few hidden layer neurons. **(B)** Same as in (A) but for Task 2.
(EPS)

**S3 Fig. Particle responsiveness metric (PRM) shows correspondence between type of training and particles preferred by the network. (A-D)** Mean and standard deviation (blue bars and black lines respectively) of the PRM for various types of training and particle orientations across ten trials. The title of each plot reflects the most recently trained stage, the vertical axis corresponds to the value of the PRM while the horizontal axis identifies the particle type (bold labels indicate ideal particles the network would be attracted to following the corresponding training). It can be seen that the metric indicates the network is most responsive to the corresponding ideal particle types following a specific training regime e.g. Post Task 1 the network is most responsive to horizontal particles (A), Post Task 2 the network is most responsive to vertical particles (B), Post $I_{S,T1}$ the network is most responsive to horizontal and vertical particles (C), Post $I_{T1,T2}$ the network is most responsive to horizontal and vertical particles (D).
(EPS)

**S4 Fig. Effect of sleep to protect old memory does not depend on specific properties of noise applied during sleep phase. (A)** Mean performance (red line) and standard deviation (blue lines) over time: unsupervised training (white), Interleaved$_{S,T1}$ (grey), Task 1/2 testing (green/yellow). **(B)** Mean and standard deviation of performance during testing on Task 1 (blue) and Task 2 (red). Following Interleaved$_{S,T1}$, mean performance on Task 1 was $0.60 \pm 0.03$ while Task 2 was $0.49 \pm 0.05$. (In all experiments, 0.5 represents chance performance.) Note that periods of Task 1 training interleaved with sleep do not lead to increase in performance on untrained Task 2, even when Task 2 data from another experiment were used to set up mean firing rates of the random input during sleep. **(C)** Same as in (A) but the sequence of training was: unsupervised training (white), Task 1 training (blue), Task 1/2 testing (green/yellow), Interleaved$_{S,T1}$ (grey), Task 1/2 testing (green/yellow). **(D)** Mean and standard deviation of performance during testing on Task 1 (blue) and Task 2 (red) after Task 1 training and after Interleaved$_{S,T1}$. Following Task 1 training, mean performance on Task 1 was $0.70 \pm 0.02$ while Task 2 was $0.53 \pm 0.02$. Post Interleaved$_{S,T1}$ training, mean performance on Task 1 was $0.71 \pm 0.02$ and Task 2 was $0.51 \pm 0.02$. Task 1 performance remained high after Interleaved$_{S,T1}$ but no improvement on Task 2 was observed. **(E)** Mean performance (red line) and standard deviation (blue lines) over time: unsupervised training (white), Task 1 training (blue), Task 1/2 testing (green/yellow), Interleaved$_{US,T2}$ (burnt orange), Task 1/2 testing (green/yellow). **(F)** Mean and standard deviation of performance during testing on Task 1 (blue) and Task 2 (red). Following Task 1 training, mean performance on Task 1 was $0.70 \pm 0.02$ while Task 2 was $0.53 \pm 0.02$. Post Interleaved$_{US,T2}$ training, mean performance on Task 1 was $0.67 \pm 0.05$ and Task 2 was $0.69 \pm 0.03$.
(EPS)

**S5 Fig. Interleaving old and new task training allows integrating synaptic information relevant to new task while preserving old task information. (A)** Mean performance (red line) and standard deviation (blue lines) over time: unsupervised training (white), Task 1 training (blue), Task 1/2 testing (green/yellow), Task 2 training (red), Task 1/2 testing (green/yellow), Interleaved$_{T1,T2}$ training (purple), Task 1/2 testing (green/yellow). **(B)** Mean and standard deviation of performance during testing on Task 1 (blue) and Task 2 (red). Following Task 1training, mean performance on Task 1 was $0.69 \pm 0.02$ while Task 2 was $0.53 \pm 0.02$. Conversely, following Task 2 training, mean performance on Task 1 was $0.52 \pm 0.02$ while Task2 was $0.69 \pm 0.04$. Following Interleaved$_{T1,T2}$ training, mean performance on Task 1 was $0.65 \pm 0.03$ while Task 2 was $0.67 \pm 0.04$. **(C)** Distributions of task-relevant synaptic weights (blue bars–single trial, orange line / shaded region–mean / std across 10 trails. The distributional structure of Task 1-relevant synapses following Task 1 training (top-left) is destroyed following Task 2 training (top-middle), but partially recovered following. Interleaved$_{T1,T2}$ training (top-right). Similarly, the distributional structure of Task 2-relevantsynapses following Task 2 training (bottom-middle), which was not present following Task 1training (bottom-left), was partially preserved following Interleaved$_{T1,T2}$ training (bottom-right).**(D)** Box plots with mean (dashed green line) and median (dashed orange line) of the distance to the decision boundary found by an SVM trained to classify Task 1 and Task 2 synaptic weight matrices for Task 1, Task 2, and Interleaved$_{T1,T2}$ training across trials. Task 1 and Task 2synaptic weight matrices had mean classification values of -0.069 and 0.069 respectively, while that of Interleaved$_{T1,T2}$ training was 0.016. **(E)** Trajectory of H to O layer synaptic weights through PC space. Synaptic weights which evolved during Interleaved$_{T1,T2}$ training (green dots)clustered in a location of PC space intermediary between the clusters of synaptic weights which evolved during training on Task 1 (red dots) and Task 2 (blue dots).
(EPS)

**S6 Fig. Freezing a fraction of task specific strong synapses preserves differing degrees of performance in a sequential learning paradigm. (A-C)** Mean and standard deviation of performance during testing on Task 1 (blue) and Task 2 (red). Left, Performance after Task 1 training. Right, Performance after Task 2 training when a fraction of the strongest (after Task 1 training) synapses remained frozen– 1% (A), 5% (B), 10% (C). In all cases, after Task 1 training, Task 1 performance was 0.70 ± 0.02 and Task 2 performance was 0.53 ± 0.02. (A) Freezing the top 1% of Task 1 synapses resulted in a Task 1 performance of 0.54 ± 0.02 and Task 2 performance of 0.68 ± 0.03. (B) Freezing the top 5% of Task 1 synapses resulted in a Task 1 performance of 0.65 ± 0.02 and Task 2 performance of 0.61 ± 0.01. (C) Freezing the top 10% of Task 1 synapses resulted in a Task 1 performance of 0.70 ± 0.03 and Task 2 performance of 0.53 ± 0.03. Freezing the top 1% of Task 1 synapses was not sufficient to maintain Task 1 performance, thus enabling Task 2 relevant synapses to dominate the network; however, freezing the top 10% of Task 1 synapses fully retains Task 1 performance preventing Task 2 to be learned.
(EPS)

## Author Contributions

**Conceptualization:** Ryan Golden, Pavel Sanda, Maxim Bazhenov.

**Data curation:** Jean Erik Delanois.

**Formal analysis:** Ryan Golden, Jean Erik Delanois, Maxim Bazhenov.

**Funding acquisition:** Maxim Bazhenov.

**Investigation:** Jean Erik Delanois.

**Methodology:** Ryan Golden, Jean Erik Delanois, Pavel Sanda, Maxim Bazhenov.

**Project administration:** Maxim Bazhenov.

**Resources:** Maxim Bazhenov.

**Software:** Jean Erik Delanois.

**Supervision:** Pavel Sanda, Maxim Bazhenov.

**Visualization:** Jean Erik Delanois.

**Writing – original draft:** Ryan Golden, Jean Erik Delanois, Pavel Sanda, Maxim Bazhenov.

**Writing – review & editing:** Ryan Golden, Jean Erik Delanois, Pavel Sanda, Maxim Bazhenov.

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
