## [Decision Letter · Decision Letter 0]

31 May 2022

Dear Dr. Bazhenov,

Thank you very much for submitting your manuscript "Sleep prevents catastrophic forgetting in spiking neural networks by forming a joint synaptic weight representation" for consideration at PLOS Computational Biology. As with all papers reviewed by the journal, your manuscript was reviewed by members of the editorial board and by several independent reviewers. The reviewers appreciated the attention to an important topic. Based on the reviews, we are likely to accept this manuscript for publication, providing that you modify the manuscript according to the review recommendations.

In particular, it would be useful to demonstrate how robust these results are; further describe and explore the variability in performance across different runs; and illustrate changes in network activity and synaptic weights during simulated wake and sleep periods, to give a better insight into the mechanism at play.

Sincerely,

Daniel Bush

Associate Editor

PLOS Computational Biology

Kim Blackwell

Deputy Editor

PLOS Computational Biology

[LINK]

In particular, it would be useful to demonstrate how robust these results are; further describe and explore the variability in performance across different runs; and illustrate changes in network activity and synaptic weights during simulated wake and sleep periods, to give a better insight into the mechanism at play

Reviewer's Responses to Questions

**Comments to the Authors:**

Reviewer #1: This is a thorough investigation of a simple artificial neural network trained sequentially on two foraging-style reinforcement learning tasks. The authors investigate in detail how the well-known phenomenon of catastrophic forgetting emerges from changes in synaptic weights during learning of one task followed by the other, and how this can be mitigated by interleaved learning, spontaneous activity similar to sleep, and freezing weights. They find that interleaving new learning with “sleep” (in which random spiking in the hidden layer triggers plasticity with an unsupervised spike-time-dependent plasticity rule) pushes the synaptic weights towards a configuration which represents the intersection of the manifolds for each task, employing several analytical tools for classification and dimensionality reduction to support this conclusion.

Catastrophic forgetting continues to be an area of high interest for researchers in the fields of neuroscience and artificial intelligence, and the insight in this manuscript of how sleep might rescue catastrophic forgetting is relevant to the field. Although neither the hypothesis nor the model and task used to test it are novel (for example references 41 and 14 by some of the same authors), they are are thorough in examining how the dynamics of synaptic weight changes evolve over simulated learning and sleep. Without directly making predictions that could be tested in brains experimentally, they add to a growing perspective in computational neuroscience about how sleep impacts learning representations, and the results could be used to inspire new experiments. It is limited in scope but thoroughly explored.

My main concern is a substantial proportion of the results are examples from a single experiment and sometimes a single neuron, with no suggestion of how well these findings generalise to other runs with other random seeds or initialisations. The manuscript would benefit from some indication of how robust these results are. For example, constructing a single metric to summarise the bias of the receptive fields in the output layer neurons towards the task-specific orientations (figure 2), and showing a distribution of these values across neurons and experiments rather than relying on a single example. This is particularly important if the authors do not intend to publicly share their code.

Some other comments:

1. Fig 1A says “apply to all middle-to-output connections”; what does middle mean here? If middle means hidden layer, better to say “hidden” for consistency and clarity. It is also not clear to me whether the blue and grey are supposed to represent different things, e.g. the blue lines next to “apply to all middle-to-output connections” suggests it is a legend indicating where the global reward signal is applied to; but a lot of the parts in blue do not receive a reward signal.

2. Lines 124-125: if I understand correctly from this, performance of 0.7 means 70% of the pairs of particles that the model encountered were “rewarded” and 30% were “punished”. I am not sure this is explained – would be helpful if this was made explicit.

3. Lines 139-140 state a standard deviation of 0.2, which is both surprisingly high and inconsistent with what is shown in figure 1; I believe this is a typo which should say 0.02

4. Fig 1 and text on lines 139,-140, 173-174, 179-180, & 199-201 mention mean and standard deviation of performance; presumably this is over a number of experiments with different random seeds, but this is not made explicit. Please indicate the number of experiments (or otherwise, what is being averaged over).

5. In fig 3E there is some interesting variability in the performance of different runs. At best (mean + standard deviation), task 1 retesting shows no forgetting at all, while at worst (mean - standard deviation) there is some forgetting with a decrement in performance of about a quarter. It would be useful for the authors to address why this might be. Perhaps in some runs the synaptic weight configurations were more orthogonal, so harder to find an intersection during interleaved training?

6. In relation to lines 431-436, finding convergent evidence in the literature for the impact of sleep on perceptual learning: these studies (reference 46, in particular) show that sleep is necessary for the initial learning of a task, which appears to be quite different from the conclusion of this manuscript that acquisition of one task does not require sleep, but sequential acquisition of two tasks does. It would be helpful if the authors could be clearer about how they believe their model relates to such findings in this respect.

Reviewer #2: How animals learn to survive and function efficiently under a variety of different conditions (and how we might be able to design artificial agents that are able to replicate this feat) is an important open question. As agents have limited resources for storing and processing information, a fundamental problem is how to dedicate resources to learning a new task while retaining useful information about previously learned tasks. Learning systems based on distributed representations and storage often lose the ability to perform previously learned tasks when learning a new task, a problem known as catastrophic forgetting (or catastrophic interference). Interleaved learning and off-line rehearsal have been suggested as solutions to this problem. The current manuscript examines this issue using simulations of an agent learning two different versions of a foraging task using a multi-layer spiking neural network that utilizes different (reward-dependent and reward-independent) forms of synaptic plasticity.

The current work builds upon a long series of earlier models developed in the laboratory of the last author. The previous papers described and analyzed in detail most components of the current model, including the (slightly unusual) single cell model, the network architecture, the reward-based learning rule and the simulated foraging task - and one of the papers showed how this artificial agent can successfully learn to obtain reward in a fixed environment. The novel contribution of the current manuscript is that the authors now examine whether the agent can successfully learn two different versions of the task that are presented sequentially. They confirm (as might be expected from earlier studies) that catastrophic forgetting also occurs in the baseline version of their architecture, and then proceed to demonstrate that a form of rehearsal (which is suggested to be analogous to processes taking place during REM sleep) can prevent catastrophic forgetting and allows the sequential acquisition of the two tasks.

In general, the results are presented in a clear and concise manner, and support the majority of the conclusions in a convincing way. One minor issue that I had was about the interpretation of the results in terms of synaptic weight changes. In principle, it could help intuitive understanding that analysis and explanations are provided both at the level of (the statistics of) single weights and in high-dimensional weight space. However, while I like the explanation of the results in terms of solution manifolds, I find the explanations in terms of single weights a bit simplistic and possibly misleading. In particular, evolution of the weight vector over one of the solution manifolds towards the intersection of the two manifolds does not necessarily entail the preservation of individual strong weights. In fact, if the preservation of strong weights is a key feature of interference-free learning in this network, then it is not clear why the simulation experiment where strong weights were explicitly frozen did not achieve the same goal - although this might depend in further details of training that are not highlighted in the paper.

Although the high-level results achieved by the agent are quite convincing, I would have been interested in learning more about what is going on "under the hood". In particular, none of the figures shows any examples of activity in the network either at the single neuron or at the population level (e.g., voltage traces or spike rasters). I think it would be useful to see illustrations of network activity (and perhaps related synaptic changes) in both the awake and the sleep phase of training.

Some further minor comments are as follows:

- The figures show time in units of "aeons". Beside the fact the this word normally indicates extremely long periods of time (in some contexts, a billion years), this unit is not defined in the Methods section, and it is not clear what it corresponds to (either in simulation time steps or in real time).

- I think that the graphical representation of the solution manifolds in Figure 7 is a bit misleading. In particular, as the solution manifolds of the individual tasks are depicted as 2D surfaces, their intersection should be a 1D (closed) curve not a 2D region.

- Have you tried to estimate the dimensionality of the solution manifolds? If so, does this provide a hint about the capacity of the network for learning multiple tasks?

- In line 548, there is a term (1-R+2XR) which does not appear in the equations, and may correspond to an alternative formulation of noise in which X is sampled from [0,1] rather than [-1,1].

- In lines 650 and 653, M should probably be M_\\tau

- The description of synaptic interactions in the Methods is a bit unclear. In particular, how is g_syn related to the W's? And what is the temporal evolution of WI_i,j? Is it subject to plasticity?

- There are several typos and grammatical errors that should be corrected.

**Have the authors made all data and (if applicable) computational code underlying the findings in their manuscript fully available?**

Reviewer #1: **No: **Manuscript states: All relevant data was simulated and can be regenerated using custom code which is

available from the corresponding author (M.B) upon request.

Reviewer #2: **No: **The source code used in this modeling study has not been made publicly available - however, the authors state that it is available from the corresponding author upon request.

PLOS authors have the option to publish the peer review history of their article (what does this mean?). If published, this will include your full peer review and any attached files.

Reviewer #1: No

Reviewer #2: **Yes: **Szabolcs Káli

Figure Files:

Data Requirements:

Reproducibility:

References:

---

## [Decision Letter · Decision Letter 1]

24 Sep 2022

Dear Dr. Bazhenov,

Thank you very much for submitting your manuscript "Sleep prevents catastrophic forgetting in spiking neural networks by forming a joint synaptic weight representation" for consideration at PLOS Computational Biology. As with all papers reviewed by the journal, your manuscript was reviewed by members of the editorial board and by several independent reviewers. The reviewers appreciated the attention to an important topic. Based on the reviews, we are likely to accept this manuscript for publication, providing that you modify the manuscript according to the review recommendations. Specifically, you must address the two minor outstanding concerns raised by Reviewer #1. 

Sincerely,

Daniel Bush

Academic Editor

PLOS Computational Biology

Kim Blackwell

Section Editor

PLOS Computational Biology

[LINK]

Reviewer's Responses to Questions

**Comments to the Authors:**

Reviewer #1: Thank you to the authors for considering my suggestions so carefully. In particular, the clarity about the number of trials with different seeds and the construction of a particle responsiveness metric make the results much easier to appreciate. My curiosity about the variability in figure 3E remains, but if the authors consider such questions to be beyond the scope of the manuscript then such is their right. Ultimately, this is a robust and convincing set of results about the role of sleep in continual learning.

I have no outstanding major concerns, but two small errors that should be addressed before publication.

1. The text regarding the particle responsiveness metric refers to figure S5, when the relevant figure is S3.

2. The sentence on line 63 of supplementary figure legends appears to end prematurely.

Reviewer #2: The authors have fully addressed all of my earlier concerns, and I have no further comments or questions.

(Note: Most of the typos and grammatical errors have been corrected, but a few minor mistakes remain: some missing articles, "It worth mentioning..." -> "It is worth mentioning...", etc.)

**Have the authors made all data and (if applicable) computational code underlying the findings in their manuscript fully available?**

Reviewer #1: **No: **Manuscript states data and code will be available upon request

Reviewer #2: **No: **The authors have not made their code publicly available (as part of the supporting information or in a public repository), although they state that the code is "available from the corresponding author (M.B) upon request."

PLOS authors have the option to publish the peer review history of their article (what does this mean?). If published, this will include your full peer review and any attached files.

Reviewer #1: No

Reviewer #2: **Yes: **Szabolcs Káli

Figure Files:

Data Requirements:

Reproducibility:

References:

---

## [Editor Report · Decision Letter 2]

3 Oct 2022

Dear Dr. Bazhenov,

We are pleased to inform you that your manuscript 'Sleep prevents catastrophic forgetting in spiking neural networks by forming a joint synaptic weight representation' has been provisionally accepted for publication in PLOS Computational Biology.

Best regards,

Daniel Bush

Academic Editor

PLOS Computational Biology

Kim Blackwell

Section Editor

PLOS Computational Biology

---

## [Editor Report · Acceptance letter]

18 Oct 2022

PCOMPBIOL-D-22-00638R2 

Sleep prevents catastrophic forgetting in spiking neural networks by forming a joint synaptic weight representation

Dear Dr Bazhenov,

I am pleased to inform you that your manuscript has been formally accepted for publication in PLOS Computational Biology. Your manuscript is now with our production department and you will be notified of the publication date in due course.

With kind regards,

Zsofia Freund
